# Adversarial Unlearning: Reducing Confidence Along Adversarial Directions

Amrith Setlur[1,*]    Benjamin Eysenbach[1]    Virginia Smith[1]    Sergey Levine[2]

[1] Carnegie Mellon University    [2] UC Berkeley

## Abstract

Supervised learning methods trained with maximum likelihood objectives often overfit on training data. Most regularizers that prevent overfitting look to increase confidence on additional examples (e.g., data augmentation, adversarial training), or reduce it on training data (e.g., label smoothing). In this work we propose a complementary regularization strategy that reduces confidence on self-generated examples. The method, which we call RCAD (Reducing Confidence along Adversarial Directions), aims to reduce confidence on out-of-distribution examples lying along directions adversarially chosen to increase training loss. In contrast to adversarial training, RCAD does not try to robustify the model to output the original label, but rather regularizes it to have reduced confidence on points generated using much larger perturbations than in conventional adversarial training. RCAD can be easily integrated into training pipelines with a few lines of code. Despite its simplicity, we find on many classification benchmarks that RCAD can be added to existing techniques (e.g., label smoothing, MixUp training) to increase test accuracy by 1–3% in absolute value, with more significant gains in the low data regime. We also provide a theoretical analysis that helps to explain these benefits in simplified settings, showing that RCAD can provably help the model unlearn spurious features in the training data.

## 1 Introduction

Supervised learning techniques typically consider training models to make accurate predictions on fresh test examples drawn from the same distribution as training data. Unfortunately, it is well known that maximizing the likelihood of the training data alone may result in overfitting. Prior work broadly considers two approaches to combat this issue. Some methods train on additional examples, e.g., generated via augmentations [53, 10, 66] or adversarial updates [14, 36, 4]. Others modify the objective by using alternative losses and/or regularization terms (e.g., label smoothing [56, 39], MixUp [69], robust objectives [65, 22]). In effect, these prior approaches either make the model's predictions more certain on new training examples or make the distribution over potential models less certain.

Existing regularization methods can be seen as providing a certain inductive bias for the model, e.g., the model's weights should be small (i.e., weight decay), the model's predictions should vary linearly between training examples (i.e., MixUp). In this paper we identify a different inductive bias: the model's predictions should be less confident on out-of-distribution inputs that look nothing like the training examples. We turn this form of inductive bias into a simple regularizer, whose benefits are complementary to existing regularization strategies. To instantiate such a method, we must be able to sample out-of-distribution examples. For this, we propose a simple approach: generating adversarial examples [38, 14] using very large step sizes (orders-of-magnitude larger than traditional adversarial

---

*Correspondence can be sent to asetlur@cs.cmu.edu.

36th Conference on Neural Information Processing Systems (NeurIPS 2022).

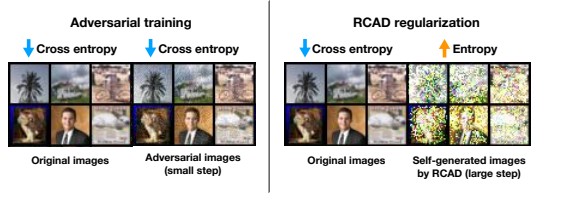
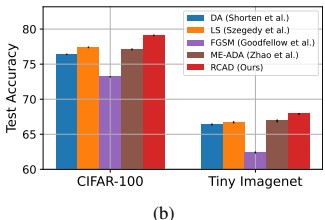

Figure 1: **Reducing confidence along adversarial directions** (RCAD) is a simple and efficient regularization technique to improve test performance. *(Left)* For RCAD, examples are generated by taking a large step (10× typical for adversarial examples) along the gradient direction. We see that generated images thus look very different from the original, with accentuated spurious components responsible for the model's flipped predictions on adversarial images. *(Right)* RCAD achieves greater test accuracy than data augmentation (DA), label smoothing (LS), and methods that minimize cross-entropy on adversarial examples: adversarial training via FGSM [14] and ME-ADA [73].

training [36]). Hence, we first perturb the training points using the training loss gradient, and then maximize predictive entropy on these adversarially perturbed examples.

In contrast to adversarial training, our method does not try to robustify the model to output the original label, but rather regularizes it to have reduced confidence on examples generated via a large step size (Figure 1a). As shown in Figure 1b, this can lead to significant improvements in in-distribution test accuracy unlike adversarial training [36, 37, 5, 4], which tends to decrease in-distribution test performance [48, 70, 61]. Compared with semi-supervised learning methods [15, 64], our method does not require an additional unlabeled dataset (it generates one automatically), and different from prior works [64, 74] it also trains the model to be less confident on these examples.

As the self-generated samples in our procedure no longer resemble in-distribution data, we are uncertain of their labels and train the model to predict a uniform distribution over labels on them, following the principle of maximum entropy [25]. A few prior works [56, 43, 52, 11] have also considered using the same principle to prevent the memorization of training examples, but only increase entropy on *iid* sampled *in-distribution* labeled/unlabeled data. In contrast, we study the effect of maximizing entropy on self-generated *out-of-distribution* examples along adversarial directions.

The main contribution of this work is a training procedure we call *RCAD: Reducing Confidence along Adversarial Directions*. RCAD improves test accuracy (for classification) and log-likelihood (for regression) across multiple supervised learning benchmarks. Importantly, we find that the benefits of RCAD are complementary to prior methods: Combining RCAD with alternative regularizers (e.g., augmentation [53, 66], label smoothing [56], MixUp training [69]) further improves performance. Our method requires adding ∼5 lines of code and is computationally efficient (with training time at most 1.3× standard). We provide a theoretical analysis that helps to explain RCAD's benefits in a simplified setting, showing that RCAD can unlearn spurious features in the training data, thereby improving accuracy on unseen examples.

## 2 Related Work

Below we survey common regularization techniques in machine learning, as well as other methods that utilize entropy maximization in training for various purposes (differing from our own).

**Data augmentation.** A common strategy to improve test accuracy (particularly on image tasks) is to augment the training data with corruptions [53] or surface level variations [10, 66] (e.g., rotations, random crops). Some methods [37, 36, 14] further augment data with imperceptibly different adversarial examples or interpolated samples [69]. Others [46, 42] sample new examples from generative models that model the marginal density of the input distribution. Also related are semi-supervised methods, which assume access to an extra set of unlabeled data and train the model to have more confident predictions on them [15, 37]. All these methods *minimize entropy* [16] on the augmented or unlabeled data to improve generalization [69, 73]. In contrast, our method *maximizes entropy* on perturbed samples along the adversarial direction.

Table 1: **Regularization objectives**: We summarize prior works that employ adversarial examples or directly regularize model's predictions $p_{\boldsymbol{w}}(\mathrm{y} \mid \mathbf{x})$ along with the scalar hyperparameters (in $[\cdot]$) associated with each.

| name | objective |
|---|---|
| cross entropy | $\min_{\boldsymbol{w}} - \sum_{\mathbf{x}, \mathrm{y} \in \hat{\mathcal{D}}} \log p_{\boldsymbol{w}}(\mathrm{y} \mid \mathbf{x})$ |
| label smoothing $[\epsilon]$ [39] | $\min_{\boldsymbol{w}} - \sum_{\mathbf{x}, \mathrm{y} \in \hat{\mathcal{D}}} \left( (1-\epsilon) \log p_{\boldsymbol{w}}(\mathrm{y} \mid \mathbf{x}) + \sum_{\mathrm{y}' \neq \mathrm{y}} \frac{\epsilon}{|\mathcal{Y}|-1} \log p_{\boldsymbol{w}}(\mathrm{y}' \mid \mathbf{x}) \right)$ |
| Adv. training $[\alpha]$ [36] | $\min_{\boldsymbol{w}} - \sum_{\mathbf{x}, \mathrm{y} \in \hat{\mathcal{D}}} \log p_{\boldsymbol{w}}(\mathrm{y} \mid \mathbf{x} - \alpha \cdot \mathrm{sign}(\nabla_{\mathbf{x}} \log p_{\boldsymbol{w}}(\mathrm{y} \mid \mathbf{x})))$ |
| ME-ADA $[\alpha, \beta]$ [73] | $\min_{\boldsymbol{w}} - \sum_{(\mathbf{x}, \mathrm{y}) \in \hat{\mathcal{D}} \cup \hat{\mathcal{D}}'} \log p_{\boldsymbol{w}}(\mathrm{y} \mid \mathbf{x})$ where, for a distance metric $C_{\boldsymbol{w}}{:}(\mathcal{X} \times \mathcal{Y}) \times (\mathcal{X} \times \mathcal{Y}) \mapsto \mathbb{R}$ 
 $\hat{\mathcal{D}}' \triangleq \{ (\tilde{\mathbf{x}}, \mathrm{y}) \mid \tilde{\mathbf{x}} \triangleq \sup_{\boldsymbol{x}_0} - \log p_{\boldsymbol{w}}(\boldsymbol{x}_0 \mid \mathrm{y}) + \alpha \mathcal{H}_{\boldsymbol{w}}(\boldsymbol{x}_0) - \beta C_{\boldsymbol{w}}((\boldsymbol{x}_0, \mathrm{y}), (\mathbf{x}, \mathrm{y})), \forall (\mathbf{x}, \mathrm{y}) \in \hat{\mathcal{D}} \}$ |
| RCAD (ours) $[\alpha, \lambda]$ | $\min_{\boldsymbol{w}} \sum_{\mathbf{x}, \mathrm{y} \in \hat{\mathcal{D}}} \left( -\log p_{\boldsymbol{w}}(\mathrm{y} \mid \mathbf{x}) - \lambda \cdot \mathcal{H}_{\boldsymbol{w}}(\mathbf{x} - \alpha \cdot \nabla_{\mathbf{x}} \log p_{\boldsymbol{w}}(\mathrm{y} \mid \mathbf{x})) \right)$ |

**Adversarial training.** Deep learning models are vulnerable to worst-case perturbations that can flip the model's predictions [14, 50]. Adversarial training was proposed to improve robustness to such attacks by reducing worst-case loss in small regions around the training samples [36, 33, 48]. More recently, Zhao et al. [73] proposed maximum entropy adversarial data augmentation (ME-ADA), which uses an information bottleneck principle to identify worst-case perturbations that both maximize training loss and predictive entropy. Similar to adversarial training ME-ADA still minimizes cross-entropy to output the same label on new points. There are two key differences between above methods and RCAD; *(i)* rather than minimizing cross entropy loss on adversarial examples, we *increase* model's uncertainty on the self-generated examples; and *(ii)* we take much *larger* steps—so large that unlike adversarial images, the generated example is no longer similar to the original one (Figure 1a). While prior work has successfully used adversarial examples to improve OOD detection [2] or adversarial robustness [36, 33, 48], we show that these changes allow our method to improve the standard test accuracy, a metric that adversarial training typically makes worse [48, 60, 70]. Further, RCAD has a lower (at most $1/5^{\mathrm{th}}$) computational cost compared to multi-step adversarial training procedures [73, 70].

**Robust objectives.** In order to improve robustness to noise and outliers in the data, a common approach is to modify the objective by considering risk averse loss functions [e.g., 22, 49, 57] or incorporating regularization terms such as the $l_2/l_1$ norms [31, 59]. Our method is similar to these approaches in that we propose a new regularization term. However, whereas most regularization terms are applied to the model's weights or activations, ours is directly applied to the model's predictions. Another effective loss function for classification problems is label smoothing [56, 8, 26], which uniformly increases model's predictive uncertainty in a region around training samples [12]. In contrast, RCAD increases entropy only on examples generated along the adversarial direction that has been shown to comprise of spurious features [23, 7], thereby *unlearning* them.

**Entropy maximization.** Finally, we note that our work builds upon prior work that draws connections between entropy maximization in supervised [52, 11, 43] and reinforcement learning [17]. For example, Pereyra et al. [43] apply a hinge form of confidence penalty directly on training data which is similar to label smoothing in principle and performance (Table 2 in [43]). Other works like [11, 52] also adapt the principle of entropy maximization but do so either on additional unlabeled data or a subset of the training samples. More recently [45] show that maximizing entropy on interpolated samples from the same class improves out-of-distribution uncertainty quantification. While we also minimize cross-entropy loss on training data, in contrast to the above we maximize entropy on samples generated along the adversarial direction. Our experimental results also span a wider set of benchmarks and presents significant gains $(+1\text{–}3\%)$ complementary to methods like label smoothing. We also theoretically analyze our objective for the class of linear predictors and show how RCAD can mitigate vulnerability to spurious correlations.

## 3 RCAD: Reducing Confidence Along Adversarial Directions

We now introduce our regularization technique for reducing confidence along adversarial directions (RCAD). This section describes the objective and an algorithm for optimizing it; Section 5 presents a more formal discussion of RCAD and in Section 4 we provide our empirical study.

**Notation.** We are given a *training* dataset $\hat{\mathcal{D}} \triangleq \big\{ (\mathbf{x}^{(i)}, \mathrm{y}^{(i)}) \big\}_{i=1}^{N}$ where $\mathbf{x}^{(i)} \in \mathcal{X}$, $\mathrm{y}^i \in \mathcal{Y}$, are sampled *iid* from a joint distribution $\mathcal{D}$ over $\mathcal{X} \times \mathcal{Y}$. We use $\hat{\mathcal{D}}$ to denote both the training dataset and

an empirical measure over it. The aim is to learn a parameterized distribution $p_{\boldsymbol{w}}(\mathrm{y} \mid \mathbf{x})$, $\boldsymbol{w} \in \mathcal{W}$ where the learnt model is typically obtained by maximizing the log-likelihood over $\hat{\mathcal{D}}$. Such a solution is referred to as the maximum likelihood estimate (MLE): $\hat{\boldsymbol{w}}_{\mathrm{mle}} \triangleq \arg\max_{\boldsymbol{w} \in \mathcal{W}} \mathbb{E}_{\hat{\mathcal{D}}} \log p_{\boldsymbol{w}}(\mathrm{y} \mid \mathbf{x})$. In the classification setting, we measure the performance of any learned solution $\hat{\boldsymbol{w}}$ using its accuracy on the *test* (population) data: $\mathbb{E}_{\mathcal{D}} \left[ \mathbb{1}(\arg\max_{y'} p_{\hat{\boldsymbol{w}}}(y' \mid \mathbf{x}) = \mathrm{y}) \right]$.

Solely optimizing the log-likelihood over the training data can lead to poor test accuracy, since the estimate $\hat{\boldsymbol{w}}_{\mathrm{mle}}$ can overfit on noise in $\hat{\mathcal{D}}$ and fail to generalize to unseen samples. Many algorithms aim to mitigate overfitting by either designing suitable loss functions replacing the log-likelihood objective, or by augmenting the training set with additional data (see Section 2). Our main contribution is a new data-dependent regularization term that will depend not just on the model parameters but also on the training dataset. We reduce confidence on out-of-distribution samples that are obtained by perturbing the original inputs along the direction that adversarially maximizes training loss.

In the paragraphs that follow we describe the methodology and rationale behind our objective that uses the following definition of model's predictive entropy when $\mathcal{Y}$ is a discrete set:

$$\mathcal{H}_{\boldsymbol{w}}(x) \triangleq -\sum_{y \in \mathcal{Y}} p_{\boldsymbol{w}}(y \mid \mathbf{x}) \, \log p_{\boldsymbol{w}}(y \mid \mathbf{x}).$$

In cases where $\mathcal{Y}$ is continuous, for example in regression tasks, we will use the differential form of predictive entropy: $-\int_{\mathcal{Y}} p_{\boldsymbol{w}}(y \mid \mathbf{x}) \, \log p_{\boldsymbol{w}}(y \mid \mathbf{x}) \, dy$.

**Reducing Confidence Along Adversarial Directions.** The key idea behind RCAD is that models should not only make accurate predictions on the sampled data, but also make uncertain predictions on examples that are very different from training data. We use directions that adversarially maximize the training loss locally around the training points to construct these out-of-distribution examples that are different from the training data. This is mainly be-

**RCAD: Reducing Confidence along Adversarial Directions**

```
def rcad_loss(x, y, α, λ):
  loss = − model(x).log_prob(y)
  x_adv = x + α * loss.grad(x)
  entropy = model(x_adv).entropy()
  return loss - λ * entropy
```

cause adversarial directions have been known to comprise of spurious features [23] and we want to regularize the model in a way that makes it uncertain on these features. We first describe how these examples are generated, and then describe how we train the model to be less confident.

We generate an out-of-distribution example $\tilde{\mathbf{x}}$ by taking a large gradient of the MLE objective with respect to the input $\mathbf{x}$, using a step size of $\alpha > 0$:

$$\tilde{\mathbf{x}} \triangleq \mathbf{x} - \alpha \cdot \nabla_{\mathbf{x}} \log p_{\boldsymbol{w}}(\mathrm{y} \mid \mathbf{x}) \tag{1}$$

We train the model to make unconfident predictions on these self-generated examples by maximizing the model's predictive entropy. We add this entropy term $\mathcal{H}_{\boldsymbol{w}}(\tilde{\mathbf{x}})$ to the standard MLE objective, weighted by scalar $\lambda > 0$, yielding the final RCAD objective:

$$\hat{\boldsymbol{w}}_{\mathrm{rcad}} \triangleq \arg\max_{\boldsymbol{w} \in \mathcal{W}} \mathbb{E}_{\hat{\mathcal{D}}} \left[ \log p_{\boldsymbol{w}}(\mathrm{y} \mid \mathbf{x}) + \lambda \cdot \mathcal{H}_{\boldsymbol{w}}(\mathbf{x} - \alpha \cdot \nabla_{\mathbf{x}} \log p_{\boldsymbol{w}}(\mathrm{y} \mid \mathbf{x})) \right] \tag{2}$$

In adversarial training, adversarial examples are generated by solving a constrained optimization problem [36, 48]. Using a first-order approximation, the adversarial example $\tilde{\mathbf{x}}$ generated by one of the simplest solvers [14] has a closed form resembling Equation 1. We note that RCAD is different from the above form of adversarial training in two ways. First, RCAD uses a much larger step size ($10\times$ larger), so that the resulting example no longer resembles the training examples (Figure 1a). Second, whereas adversarial training updates the model to be more confident on the new example, RCAD trains the model to be less confident. Our experiments in Section 4 (Figure 4b) show that these differences are important for improving test accuracy.

**Informal understanding of RCAD.** For image classification, image features that are pure noise and independent of the true label can still be spuriously correlated with the labels for a finite set of *iid* drawn examples [54]. Such features are usually termed *spurious features* [72]. Overparameterized neural networks have a tendency to overfit on spurious features in the training examples [68, 51]. In order to generate unrealistic out-of-distribution samples, we pick examples that are far away from the true data point along the adversarial direction because *(i)* adversarial directions are comprised of noisy features that are spuriously correlated with the label on a few samples [23]; and *(ii)* maximizing entropy on samples with an amplified feature forces the model to quickly unlearn that feature. By

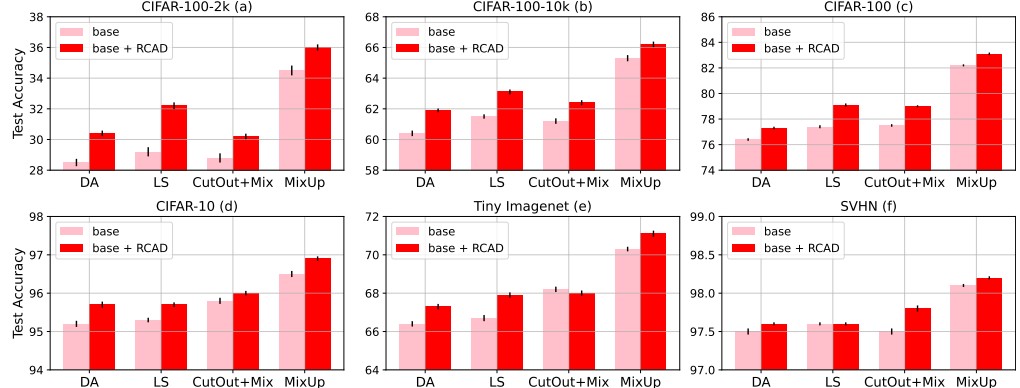

Figure 2: **Main results on supervised image classification benchmarks:** We plot the mean test accuracy and $95\%$ confidence intervals over $10$ independent runs for models trained with base methods: Data Augmentation (DA), Label Smoothing (LS), CutOut/CutMix (CutOut+Mix) augmentation, MixUp and compare them with the test accuracies of the models trained with the RCAD objective in Equation 2, in addition to the base methods.

using a large step size $\alpha$ we exacerbate the spurious features in our self-generated example $\tilde{\mathbf{x}}$. When we maximize predictive entropy over our generated examples we then force the model to *unlearn* these spurious correlations. Hence, the trained model can generalize better and achieve higher prediction performance on unseen test examples. In Section 5 we build on this informal understanding to present a more formal analysis on the benefits of RCAD.

## 4 Experiments: Does RCAD Improve Test Performance?

Our experiments aim to study the effect that RCAD has on test accuracy, both in comparison to and in addition to existing regularization techniques and adversarial methods, so as to understand the degree to which its effect is *complementary* to existing methods.

**Benchmark datasets.** We use six image classification benchmarks. In addition to CIFAR-10, CIFAR-100 [30], SVHN [41] and Tiny Imagenet [34], we modify CIFAR-100 by randomly sub-sampling 2,000 and 10,000 training examples (from the original 50,000) to create CIFAR-100-2k and CIFAR-100-10k. These smaller datasets allow us to study low-data settings where we expect the generalization gap to be larger. CIFAR-100(-2k/10k) share the same test sets. If the validation split is not provided by the benchmark, we hold out $10\%$ of our training examples for validation.

**Implementation details**[2]**.** Unless specified otherwise, we train all methods using the ResNet-18 [19] backbone, and to accelerate training loss convergence we clip gradients in the $l_2$ norm (at $1.0$) [71, 18]. We train all models for 200 epochs and use SGD with an initial learning rate of $0.1$ and Nesterov momentum of $0.9$, and decay the learning rate by a factor of $0.1$ at epochs $100, 150$ and $180$ [10]. We select the model checkpoint corresponding to the epoch with the best accuracy on validation samples as the final model representing a given training method. For all datasets (except CIFAR-100-2k and CIFAR-100-10k for which we used $32$ and $64$ respectively) the methods were trained with a batch size of $128$. For details on algorithm specific hyperparameter choices refer to Appendix B.

**Baselines.** The primary aim of our experiments is to study whether entropy maximization along the adversarial direction shrinks the generalization gap. Hence, we explore baselines commonplace in deep learning that either *(i)* directly constrain the model's predictive distribution on observed samples (label smoothing [39]) or *(ii)* implicitly regularize the model by training on additional images generated in an adversarial manner. For the first, our main comparisons are to label smoothing [56]), standard data augmentation [53], cutout data augmentation [66, 10], and MixUp [69] training. For the second, we compare with adversarial training[36] that uses FGSM [14] to perturb the inputs. Additionally, we compare RCAD with two recent approaches that use adversarial examples in different ways: adversarial data augmentation (ADA) [62] and maximum entropy adversarial data augmentation (ME-ADA) [73]. We summarize these baselines in Table 1.

---

[2]Code for this work can be found at `https://github.com/ars22/RCAD-regularizer`.

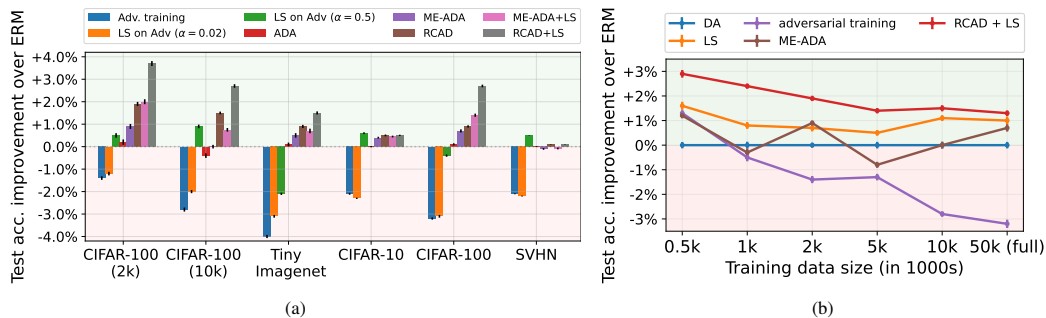

Figure 3: *(Left)* **How should we use adversarial examples to improve test accuracy?** We compare RCAD with adversarial baselines including label smoothing on adversarial samples (LS on Adv), measuring the improvement in test accuracy relative to ERM. We also compare RCAD with ME-ADA when combined with label smoothing. *(Right)* **RCAD is more effective in low-data regime.** We compare the test accuracy improvement over ERM for baselines: Label Smoothing (LS), Data Augmentation (DA), adversarial training, and ME-ADA with RCAD + LS on different sub-samples of CIFAR-100 training set (0.5k → 50k). We find RCAD achieves the largest gains in the low data regime. In both plots, we plot the mean and 95% confidence intervals over 10 independent runs.

## 4.1 How much does RCAD improve test accuracy in comparison and in addition to other regularization methods?

Figure 2 presents the main empirical findings of RCAD over six image classification benchmarks and across four baseline regularizers. Across all datasets, we observe that training with the RCAD objective improves the test accuracy over the baseline method in $22/24$ cases. The effects are more pronounced on datasets with fewer training samples. For example, on CIFAR-100-2k, adding RCAD on top of label smoothing boosts the performance by $\approx 3\%$. These results also show that the benefits of RCAD are complementary to prior methods—RCAD + MixUp outperforms MixUp, and RCAD + augmentation/smoothing outperforms both. We test for statistical significance using a 1-sided $p$-value, finding that $p \ll 1e{-}3$ in $22/24$ comparisons. In summary, these results show that our proposed regularizer is complementary to prior regularization techniques.

## 4.2 How effectively does RCAD improve test accuracy compared to adversarial training?

Our next set of experiments compares different ways of using adversarial examples. RCAD maximizes the model's predictive entropy on unlabeled examples along the adversarial direction (obtained with a large step-size $\approx 0.5$). In contrast, other methods we compare against (Adversarial Training [36], ADA [62], ME-ADA [62]) minimize the cross entropy loss on examples obtained by adversarially perturbing the inputs without changing the original labels (using a much smaller step size $\approx 0.05$). Additionally, we look at the baseline that performs label smoothing on these adversarial samples (LS on Adv) – a relaxed version of RCAD. We evaluate all methods by measuring their test accuracy improvement over empirical risk minimization (ERM), noting that some of these methods were proposed to optimize robustness, a different metric. We show results in Figure 3a and note that RCAD outperforms adversarial training and LS on Adv with small step-sizes ($\alpha = 0.02$), ADA and ME-ADA by significant margins on all benchmarks. We numerically verify that RCAD statistically outperforms the best baseline ME-ADA by computing 1-sided $p$-values ($p$=0.009 on CIFAR-10, $p < 1e{-}4$ on others).

Next, we look at LS on Adv with large $\alpha = 0.5$. Since this is the same value of $\alpha$ used by RCAD for most datasets, this method is equivalent to performing label smoothing on examples generated by RCAD. Since label smoothing is a relaxed form of entropy maximization it is not surprising that the performance of this method is similar to (if not better than) RCAD on a few benchmarks. Finally, when both RCAD and ME-ADA are equipped with label smoothing, both methods improve, but the benefit of RCAD persists ($p = 0.0478$ on CIFAR-10, $p < 1e{-}4$ on others).

## 4.3 How effective is RCAD in the low training data regime?

Since supervised learning methods are more susceptible to overfitting when training samples are limited [13], we analyze the effectiveness of RCAD in the low-data regime. To verify this, we sample

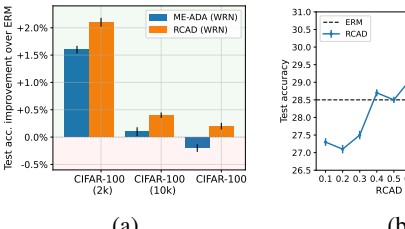

| Negative log-likelihood (NLL) on test data | | | | |
|---|---|---|---|---|
| Method | Boston | Concrete | Energy | Wine | Yacht |
| ERM | 2.85 | 3.25 | **1.85** | **0.97** | 1.66 |
| | ±0.06 | ±0.07 | ±0.03 | ±0.04 | ±0.04 |
| ME-ADA | 3.01 | **3.14** | 1.99 | 1.11 | 1.78 |
| | ±0.04 | ±0.03 | ±0.06 | ±0.05 | ±0.04 |
| RCAD (ours) | **2.60** | 3.12 | 1.75 | 1.09 | **1.54** |
| | ±0.05 | ±0.04 | ±0.07 | ±0.05 | ±0.05 |

(a)           (b)           (c)

Figure 4: **RCAD can be used with larger networks and is also effective on regression tasks.** *(Left)* We compare the improvements in test accuracy (over ERM) for RCAD and ME-ADA when all methods use Wide ResNet 28-10 [67]. *(Center)* For CIFAR-100-2k, we show the effect on test accuracy of the step size $\alpha$ that RCAD takes while generating out-of-distribution examples along the adversarial direction. *(Right)* We compare the negative log-likelihood on test samples for RCAD with baselines ERM and ME-ADA on five regression datasets from the UCI repository. For the above we show the mean and $95\%$ confidence intervals across 10 runs.

small training datasets of size 0.5k, 1k, 2k, 5k and 10k from the 50,000 training samples in CIFAR-100. The test set for each is the same as that of CIFAR-100. We train the baseline regularizers and RCAD on each of these datasets and compare the observed improvements in test accuracies relative to ERM. We show our results for this experiment in Figure 3b. Straight away we observe that RCAD yields positive improvements in test accuracy for any training dataset size. In contrast, in line with the *robust overfitting* phenomena described in Raghunathan et al. [48, 47], Zhang et al. [70], adversarial training (with FGSM [14]) is found to be hurtful in some settings. Next we observe that compared to typical regularizers for supervised learning like label smoothing and data augmentation, the regularization effect of RCAD has a stronger impact as the size of the training data reduces. Notice that RCAD has a steeper upward trend (right→left) compared to the next best method label smoothing, while outperforming each baseline in terms of absolute performance values.

## 4.4 Additional Results and Ablations

In addition to the experiments below on wider architectures, effect of step size $\alpha$, and regression tasks, we conduct more analysis and experiments for RCAD (e.g., evaluating its robustness to adversarial perturbations and distribution shifts). For these and other experimental details refer to Appendix B, C.

**Results on a larger architecture.** We compare the test accuracies of RCAD and ME-ADA (most competitive baseline from ResNet-18 experiments) when trained with the larger backbone Wide ResNet 28-10 [67] (WRN) on CIFAR-100 and its derivatives. We plot these test accuracies relative to ERM trained with WRN in Figure 4a. Clearly, the benefit of RCAD over ME-ADA still persists albeit with slightly diminished absolute performance differences compared to ResNet-18 in Figure 3a.

**Effect of step size $\alpha$ on test accuracy.** In Figure 4b we plot the test accuracy of RCAD on CIFAR-100-2k as we vary the step size $\alpha$. We see that RCAD is effective only when the step size is sufficiently large ($> 0.5$), so that new examples do not resemble the original ones. When the step size is small ($< 0.5$), the perturbations are small and the new example is indistinguishable from the original, as in adversarial training. Our theory in Section 5 also agrees with this observation.

**Results on regression tasks.** On five regression datasets from the UCI repository we compare the performance of RCAD with ERM and ME-ADA in terms of test negative log-likelihood (NLL) ($\downarrow$ is better). To be able to compute NLL, we model the predictive distribution $p_{\boldsymbol{w}}(\mathrm{y} \mid \mathbf{x})$ as a Gaussian and each method outputs two parameters (mean, variance of $p_{\boldsymbol{w}}(\mathrm{y} \mid \mathbf{x})$) at each input $\mathbf{x}$. For the RCAD regularizer we use the differential form of entropy. Results are shown in Figure 4c from which we see that RCAD matches/improves over baselines on 4/5 regression datasets.

## 5 Analysis: Why Does RCAD Improve Test Performance?

In this section, we present theoretical results that provide a more formal explanation as to why we see a boost in test accuracy when we regularize a classifier using RCAD. We analyze the simplified problem of learning $l_2$-regularized linear predictors in a fully specified binary classification setting. Specifically, our analysis will show that, while linear predictors trained with standard ERM can have

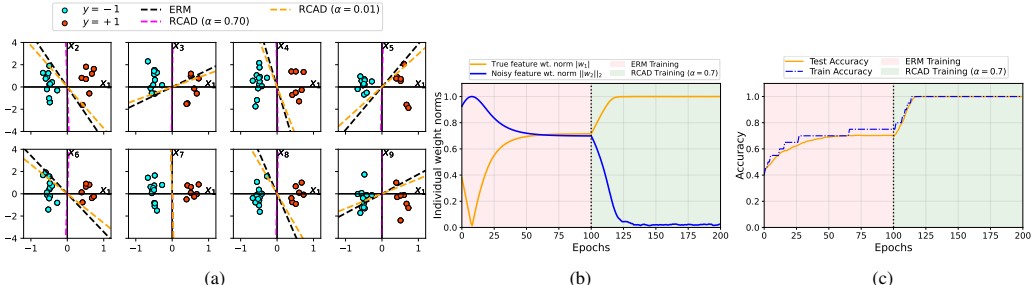

(a)            (b)           (c)

Figure 5: **RCAD corrects ERM solution:** *(Left)* We plot the training data and high-dimensional decision boundaries for all three estimates $\hat{\boldsymbol{w}}_{\mathrm{erm}}$ and $\hat{\boldsymbol{w}}_{\mathrm{rcad}}$ with $\alpha = 0.01, 0.70$. Here, we plot the linear boundary projected onto the set of planes $\{x_1, x_i\}_{i=2}^{i=9}$ spanned by the true feature $x_1$ and noisy features $x_2, \ldots, x_8$. Across the ERM and RCAD training iterations, we plot the weight norms along the true feature $|w_1|$ and noisy dimensions $\|\boldsymbol{w}_2\|_2$ in the *(Center)* plot, and on the *(Right)* we plot the train/test accuracies.

a high dependence on spurious features, further optimization with the RCAD objective will cause the classifier to unlearn these spurious features.

The intuition behind our result can be summarized as follows: when RCAD generates examples after taking a large step along adversarial directions, a majority of the generated examples end up lying close to the decision boundary along the true feature, and yet a portion of them would significantly depend on the noisy spurious features. Thus, when RCAD maximizes the predictive entropy on these new examples, the classifier weights that align with the spurious components are driven toward smaller values. Detailed proofs for the analysis that follows can be found in Appendix A.

**Setup.** We consider a binary classification problem with a joint distribution $\mathcal{D}$ over $(\mathbf{x}, \mathrm{y}) \in \mathcal{X} \times \{0, 1\}$ where $\mathcal{X} \subseteq \mathbb{R}^{d+1}$ with $\mathbf{x} = [x_1, \mathbf{x}_2]^\top$, $x_1 \in \mathbb{R}$, $\mathbf{x}_2 \in \mathbb{R}^d$ and for some $\beta > 0$, $\mathcal{D}$ is:

$$\mathrm{y} \sim \mathrm{Unif}\{-1, 1\}, \quad x_1 \sim \mathcal{N}(\beta \cdot \mathrm{y}, \sigma_1^2), \quad \mathbf{x}_2 \sim \mathcal{N}(\mathbf{0}, \Sigma_2), \quad \text{and let} \ \tilde{\Sigma} \triangleq \begin{pmatrix} \sigma_1^2 & \mathbf{0} \\ \mathbf{0} & \Sigma_2 \end{pmatrix}. \quad (3)$$

We borrow this setup from Chen et al. [7] which analyzes self-training/entropy minimization on unlabeled out-of-distribution samples. While their objective is very different from RCAD, their setup is relevant for our analysis since it captures two kinds of features: $x_1$, which is the *true* univariate feature that is predictive of the label $y$, and $\mathbf{x}_2$, which is the high dimensional *noise*, that is not correlated with the label under the true distribution $\mathcal{D}$, but w.h.p. is correlated with the label on finite sampled training dataset $\hat{\mathcal{D}} = \{(\mathbf{x}^{(i)}, \mathrm{y}^{(i)})\}_{i=1}^n \sim \mathcal{D}^n$. We wish to learn the class of homogeneous linear predictors $\mathcal{W} \triangleq \{\boldsymbol{w} \in \mathbb{R}^{d+1} : \langle \boldsymbol{w}, \mathbf{x} \rangle = w_1 \cdot x_1 + \boldsymbol{w}_2 \cdot \mathbf{x}_2, \|\boldsymbol{w}\|_2 \leqslant 1\}$. For mathematical convenience we make three modifications to the RCAD objective in Equation 2 that are common in works analyzing logistic classifiers and entropy based objectives [55, 7]: *(i)* we minimize the margin loss $l_\gamma(\boldsymbol{w}; (\mathbf{x}, \mathrm{y})) \triangleq \max(0, \gamma - \mathrm{y} \cdot \langle \boldsymbol{w}, \mathbf{x} \rangle)$ (instead of the negative log-likelihood) over $\mathcal{D}'$ with some $\gamma > 0$; *(ii)* we consider the constrained optimization problem in Equation 4 as opposed to the Lagrangian form of RCAD in Equation 2 [35]; and *(iii)* we use Lemma 5.1 and replace $\mathcal{H}_{\boldsymbol{w}}(\mathbf{x} + \alpha \cdot \nabla_{\mathbf{x}} l_\gamma(\boldsymbol{w}; (\mathbf{x}, \mathrm{y})))$ with $\exp(-|\langle \boldsymbol{w}, \mathbf{x} + \alpha \cdot \nabla_{\mathbf{x}} l_\gamma(\boldsymbol{w}; (\mathbf{x}, \mathrm{y})) \rangle|)$. Note that the optimal $\boldsymbol{w}^* \in \mathcal{W}$ that has the best test accuracy and lowest population margin loss is given by $\boldsymbol{w}^* \triangleq [1, 0, \ldots, 0]^\top$.

$$\max_{\boldsymbol{w}} \ \mathcal{M}_{\hat{\mathcal{D}}}(\boldsymbol{w}) \triangleq \mathbb{E}_{\hat{\mathcal{D}}} \ \exp(-|\langle \boldsymbol{w}, \mathbf{x} + \alpha \cdot \nabla_{\boldsymbol{x}} l_\gamma(\boldsymbol{w}; (\mathbf{x}, \mathrm{y})) \rangle|)$$
$$\text{s.t.} \quad \mathbb{E}_{\hat{\mathcal{D}}} \ \mathbb{1}(l_\gamma(\boldsymbol{w}; (\mathbf{x}, \mathrm{y})) > 0) \leqslant \rho/2, \quad \boldsymbol{w} \in \mathcal{W} \quad (4)$$

**Lemma 5.1** ([6, 55], informal). $\mathcal{H}_{\boldsymbol{w}}(\mathbf{x}) = \mathcal{H}_{\mathrm{bin}}((1 + \exp(-\langle \boldsymbol{w}, \mathbf{x} \rangle))^{-1})$ *where* $\mathcal{H}_{\mathrm{bin}}(p)$ *is the entropy of* $\mathrm{Bern}(p)$ *distribution. Thus* $\mathcal{H}_{\boldsymbol{w}}(\mathbf{x}) \approx \exp(-|\langle \boldsymbol{w}, \mathbf{x} \rangle|)$, *as both exhibit same tail behavior.*

We analyze the RCAD estimate $\hat{\boldsymbol{w}}_{\mathrm{rcad}} \triangleq \boldsymbol{w}^{(T)}$ returned after $T$ iterations of projected gradient ascent on the non-convex objective in Equation 4, initialized with $\boldsymbol{w}^{(0)}$ satisfying the constraints in Equation 4. Given the projection $\Pi_{\mathcal{S}}$ onto the convex set $\mathcal{S}$, and learning rate $\eta > 0$, the update rule for $\boldsymbol{w}^{(t)}$ is as follows: $\tilde{\boldsymbol{w}}^{(t+1)} = \boldsymbol{w}^{(t)} + \eta \cdot \nabla_{\boldsymbol{w}^{(t)}} \mathcal{M}_{\hat{\mathcal{D}}}(\boldsymbol{w})$, and $\boldsymbol{w}^{(t+1)} = \Pi_{\|\boldsymbol{w}\|_2 \leqslant 1} \tilde{\boldsymbol{w}}^{(t+1)}$.

Since the initialization $\boldsymbol{w}^{(0)}$ satisfies the constraint in Equation 4, it has a low margin loss on training samples and using margin based generalization gaps [27] we can conclude w.h.p. $\boldsymbol{w}^{(0)}$ has low test error ($\leqslant \rho$). This, in turn tells us that $\boldsymbol{w}^{(0)}$ should have learnt the true feature $x_1$ to some extent (Lemma 5.2). Intuitively, if the classifier is not trained at all, then adversarial directions would be less meaningful since they may include a higher component of the true feature $\mathbf{x}_1$, as opposed to noisy $\mathbf{x}_2$. To obtain $\boldsymbol{w}^{(0)}$, we simply minimize loss $l_\gamma$ on $\hat{\mathcal{D}}$ using projected (onto $\|\boldsymbol{w}\|_2 \leqslant 1$) gradient descent and the ERM solution $\hat{\boldsymbol{w}}_{\mathrm{erm}}$ serves as the initialization $\boldsymbol{w}^{(0)}$ for RCAD's gradient ascent iterations. Note that $\hat{\boldsymbol{w}}_{\mathrm{erm}}$ still significantly depends on spurious features (see Figure 5). Furthermore, this dependence is unavoidable and worse when $n$ is small, or when the noise dimension $d$ is large.

**Lemma 5.2** (true feature is partially learnt before RCAD). *If $\boldsymbol{w}^{(0)}$ satisfies constraint in  Equation 4, and when $n \gtrsim \frac{\beta^2 + \log(1/\delta) \cdot \|\tilde{\Sigma}\|_{\mathrm{op}} + \|\tilde{\Sigma}\|_*}{\gamma^2 \rho^2}$, with probability $1 - \delta$ over $\hat{\mathcal{D}}$,   $w_1^{(0)} \geqslant \frac{\mathrm{erfc}^{-1}(2\rho) \cdot \sqrt{2\sigma_{\min}(\tilde{\Sigma})}}{\beta}$.*

We are now ready to present our main results in Theorem 5.3 which states that after $T = \mathcal{O}(\log(1/\epsilon))$ gradient ascent iterations of optimizing the RCAD objective $\hat{\boldsymbol{w}}_{\mathrm{rcad}}$ is $\epsilon$ close to $\boldsymbol{w}^*$, since it *unlearns* the spurious feature ($\|\boldsymbol{w}_2^{(T)}\|_2 \leqslant \epsilon$) and *amplifies* the true feature ($|w_1^{(T)}| \geqslant \sqrt{1 - \epsilon^2}$). Note that Theorem 5.3 suggests a minimum step size $\alpha$ in arriving at a sufficient condition for RCAD to unlearn spuriousness and improve test performance This is also in line with our empirical findings (Figure 4b). Since $\alpha = \Theta(\gamma)$, most of the points generated by RCAD lie close to the decision boundary along $\mathbf{x}_1$, except for the ones with noisy features that are correlated with the classifier weights.

**Theorem 5.3** ($\hat{\boldsymbol{w}}_{\mathrm{rcad}} \to \boldsymbol{w}^*$). *If $\beta = \Omega(\|\tilde{\Sigma}\|_{\mathrm{op}})$ and $\exists c_0, K > 0$, such that $\beta \gtrsim \alpha \gtrsim \max(\gamma, \|\tilde{\Sigma}\|_{\mathrm{op}})$ and $\gamma + c_0 \cdot \sigma_{\min}(\tilde{\Sigma}) \geqslant \beta$, then with $n \gtrsim \frac{\beta^2 + \log(1/\delta) \cdot \|\tilde{\Sigma}\|_{\mathrm{op}} + \|\tilde{\Sigma}\|_*}{\gamma^2 \mathrm{erfc}\left(K\alpha / \sqrt{2\sigma_{\min}(\tilde{\Sigma})}\right)^2} + \frac{\log 1/\delta}{\epsilon^4}$, after $T = \mathcal{O}\left(\log(1/\epsilon)\right)$ gradient ascent iterations, $|w_1^{(T)}| \geqslant \sqrt{1 - \epsilon^2}$ and $\|\boldsymbol{w}_2^{(T)}\|_2 \leqslant \epsilon$, with probability $1 - \delta$ over $\mathcal{D}'$.*

The key idea behind our guarantee above is an inductive argument. We show that, if the class separation $\beta$ and step size $\alpha$ are sufficiently larger than any noise variance, then RCAD monotonically increases the weight norm along the true feature and monotonically decreases it along the noisy dimensions with each gradient ascent update. This is because at any given instant the gradient of RCAD objective $\mathcal{M}_{\mathcal{D}}(\boldsymbol{w})$ with respect to $w_1$ always points in the same direction, while the one with respect to $\boldsymbol{w}_2$ is sufficiently anti-correlated with the direction of $\boldsymbol{w}_2$. We formalize this argument in Lemma 5.4. It is also easy to see why the update will improve test accuracy monotonically, thus satisfying the constraint in Equation 4 with high probability.

**Lemma 5.4** ($w_1, \boldsymbol{w}_2$ update). *If $\alpha, \beta, \gamma$ and sample size $n$ satisfy the noise conditions in Theorem 5.3, then $\langle \frac{\partial \mathcal{M}_{\mathcal{D}}(\boldsymbol{w})}{\partial w_1^{(t)}}, w_1^{(t)} \rangle > 0$, and $|\tilde{w}_1^{(t+1)}| > |w_1^{(t)}|$. On the other hand, $\exists c_1 > 0$ such that $\langle \nabla_{\boldsymbol{w}_2^{(t)}} \mathcal{M}_{\mathcal{D}}(\boldsymbol{w}^{(t)}), \boldsymbol{w}_2^{(t)} \rangle \leqslant -c_1 \cdot \|\boldsymbol{w}_2^{(t)}\|_2^2$. Consequently $\exists \eta$, such that $\|\tilde{\boldsymbol{w}}_2^{(t+1)}\|_2 / \|\boldsymbol{w}_2^{(t)}\|_2 < 1$.*

**Empirically validating our theoretical results in the toy setup.** Now, using the same setup as our theoretical study, we check if taking the ERM solution and updating it with RCAD truly helps in unlearning the spurious feature $\mathbf{x}_2$ and amplifying the true feature $\mathbf{x}_1$. With $d = 8, \gamma = 2.0, \beta = 0.5, \sigma_1 = 0.1, \Sigma = \mathbf{I}_d$ we collect training dataset $\hat{\mathcal{D}}$ of size 20 according to the data distribution in Equation 3. First, we obtain the ERM solution $\hat{\boldsymbol{w}}_{\mathrm{erm}}$ by optimizing the margin loss on $\hat{\mathcal{D}}$ for 100 projected gradient descent iterations. Then, with $\hat{\boldsymbol{w}}_{\mathrm{erm}}$ as initialization we optimize the RCAD objective in Equation 4 for 100 additional projected gradient ascent iterations to obtain RCAD estimate $\hat{\boldsymbol{w}}_{\mathrm{rcad}}$. For RCAD, we try both small ($\alpha = 0.01$) and large ($\alpha = 0.7$) values of the step size. Using the results of this study in Figure 5 we shall now verify our main theoretical claims:

(i) **RCAD unlearns spurious features learnt by ERM (Theorem 5.3):** RCAD training with large step size $\alpha = 0.7$ corrects ERM's decision boundaries across all noisy dimensions i.e., $\hat{\boldsymbol{w}}_{\mathrm{rcad}}$ has almost no dependence on noisy dimensions $\{\mathbf{x}_i\}_{i=2}^{i=8}$ (Figure 5a). On the other hand, the ERM solution has a significant non-zero component across all noisy dimensions (except $\mathbf{x}_7$).

(ii) **RCAD is helpful only when step size is large enough (Theorem 5.3):** In Figure 5a we see that RCAD estimate with small step size $\alpha = 0.01$ has a higher dependence on spurious features compared to ERM. Thus optimizing RCAD with a very small $\alpha$ leads to an even poor solution than ERM. This is because, the perturbed data point still has a significant component of the true feature, which the model is forced to unlearn when RCAD tries to maximize entropy over the perturbed point.

(iii) **The weight norms $|w_1|$ and $\|w_2\|_2$ change monotonically through RCAD iterations (Lemma 5.4):** In Figure 5b we see that the norm along the true feature $|w_1|$ increases monotonically through RCAD iterations, whereas $\|\boldsymbol{w}_2\|_2$ decreases monotonically, until $|w_1| \approx 1$ and $\|\boldsymbol{w}_2\|_2 \approx 0$. Thus, RCAD successfully recovers optimal $\boldsymbol{w}^*$. Based on this, we also see the test accuracy increase monotonically through RCAD iterations; reaching $100\%$ by the end of it (Figure 5c). In contrast, since the ERM solution depends on spurious features, its accuracy does not improve beyond $70\%$ in the first 100 iterations. We find that training ERM for more iterations only improves training accuracy.

The analysis above explains why RCAD improves test performance in the linear setting and it is inline with our intuition of RCAD's ability to unlearn spurious features. To check if our intuition also generalizes to deeper networks we further study RCAD's behaviour in a non-linear setting using a different toy example in Appendix D. Furthermore, while our analysis is restricted to linear classifiers, linear models can provide a rough proxy for neural network learning dynamics via the neural tangent kernel (NTK) approximation [24]. This strategy has been used in a number of prior works [40, 58, 3] and extending the analysis to the NTK setting may be possible in future work.

## 6  Conclusion

In this paper we propose RCAD, a regularization technique that maximizes a model's predictive entropy on out-of-distribution examples. These samples lie along the directions that adversarially maximize the loss locally around the training points. Our experiments on image classification benchmarks show that RCAD not only improves test accuracy by a significant margin, but that it can also seamlessly compose with prior regularization methods. We find that RCAD is particularly effective when learning from limited data. Finally, we present analyses in a simplified setting to show that RCAD can help the model unlearn noisy features. Some current limitations of our work are that RCAD slows training by $30\%$ and our theoretical analysis is limited to the linear case, which would be interesting directions to address/expand on in future work—particularly in light of the significant empirical benefits of RCAD for improving generalization.

**Acknowledgements.** The authors would like to thank Oscar Li, Ziyu Liu, Saurabh Garg at Carnegie Mellon University and members of the RAIL lab at UC Berkeley for helpful feedback and discussion.

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
