## Appendix Outline

## A   Proofs for Section 5

In Section 5 of the main paper we analyze the performance of our method RCAD and compare it to empirical risk minimization (ERM) in a simplified Binary classification setting. Our investigation reveals that one reason for the better test performance of RCAD could be its ability to unlearn noisy features that are spuriously correlated with the labels in the finite sampled training dataset. This is identified under some conditions over the data distribution (noise conditions) and specifically when the RCAD objective is optimized using iterations of projected gradient ascent initialized with a decent (better than random) classifier. This implies that the initial classifier (initialized with the ERM solution in our case) has learnt the true feature to some extent. The solution obtained at the end of projected gradient ascent iterations on the RCAD objective maximizing entropy along adversarial directions, is termed as the RCAD solution. We also matched some of our theoretical results to trends observed empirically with respect to both the decision boundary learnt by RCAD vs. ERM classifier and the effect of the step size $\alpha$ on the performance of RCAD (Figure 5).

In this section we present proofs for our theoretical claims in Section 5. For the benefit of the reader, we begin by redefining our setup with some additional notation, followed by technical details on how the new examples are generated by RCAD in this setup. We then provide an informal proof outline for our main result in Theorem 5.3.

**Setup.**   We consider a binary classification problem with a joint distribution $\mathcal{D}$ over $(\mathbf{x}, \mathrm{y}) \in \mathcal{X} \times \{0, 1\}$ where $\mathcal{X} \subseteq \mathbb{R}^{d+1}$ with $\mathbf{x} = [\mathrm{x}_1, \mathbf{x}_2]^\top$, $\mathrm{x}_1 \in \mathbb{R}$, $\mathbf{x}_2 \in \mathbb{R}^d$ and for some $\beta > 0$, $\mathcal{D}$ is:

$$\mathrm{y} \sim \mathrm{Unif}\{-1, 1\}, \quad \mathrm{x}_1 \sim \mathcal{N}(\beta \cdot \mathrm{y}, \sigma_1^2), \quad \mathbf{x}_2 \sim \mathcal{N}(\mathbf{0}, \Sigma_2)$$

$$\tilde{\Sigma} \triangleq \begin{pmatrix} \sigma_1^2 & \mathbf{0} \\ \mathbf{0} & \Sigma_2 \end{pmatrix}, \quad \sigma_{\boldsymbol{w}} \triangleq \sqrt{\boldsymbol{w}^T \tilde{\Sigma} \boldsymbol{w}}, \quad a_{\boldsymbol{w}} \triangleq \frac{\beta w_1 - \gamma}{\sigma_{\boldsymbol{w}}} \tag{5}$$

This setup is relevant for our analysis since it captures two kinds of features: $\mathrm{x}_1$, which is the *true* univariate feature that is predictive of the label $\mathrm{y}$, and $\mathbf{x}_2$, which is the high dimensional *noise*, that is not correlated with the label under the true distribution $\mathcal{D}$, but w.h.p. is correlated with the label on finite sampled training dataset $\hat{\mathcal{D}} = \{(\mathbf{x}^{(i)}, \mathrm{y}^{(i)})\}_{i=1}^n \sim \mathcal{D}^n$. We wish to learn the class of homogeneous linear predictors $\mathcal{W} \triangleq \{\boldsymbol{w} \in \mathbb{R}^{d+1} : \langle \boldsymbol{w}, \mathbf{x} \rangle = w_1 \cdot \mathrm{x}_1 + \boldsymbol{w}_2 \cdot \mathbf{x}_2, \|\boldsymbol{w}\|_2 \leqslant 1\}$. Following the modifications we make to the RCAD objective in Section 5, and using Lemma 5.1 we arrive at the following constrained form of the RCAD objective where $l_\gamma(\boldsymbol{w}; (\mathbf{x}, \mathrm{y})) \triangleq \max(0, \gamma - \mathrm{y} \cdot \langle \boldsymbol{w}, \mathbf{x} \rangle)$.

$$\max_{\boldsymbol{w}} \quad \mathcal{M}_{\hat{\mathcal{D}}}(\boldsymbol{w}) \triangleq \mathbb{E}_{\hat{\mathcal{D}}} \ \exp(-|\langle \boldsymbol{w}, \mathbf{x} + \alpha \cdot \nabla_{\boldsymbol{x}} l_\gamma(\boldsymbol{w}; (\mathbf{x}, \mathrm{y})) \rangle|)$$

$$\mathrm{s.t.} \quad \mathbb{E}_{\hat{\mathcal{D}}} \ \mathbb{1}(l_\gamma(\boldsymbol{w}; (\mathbf{x}, \mathrm{y})) > 0) \leqslant \rho/2, \quad \boldsymbol{w} \in \mathcal{W} \tag{6}$$

Note that the gradient $\nabla_{\mathbf{x}} l_\gamma(\boldsymbol{w}; (\mathbf{x}, \mathrm{y}))$ maybe undefined at the margins, when $y \cdot \langle \boldsymbol{w}, \mathbf{x} \rangle = \gamma$. Hence, we rely on subgradients defined in Equation 7. Since, many subgradient directions exist for the margin points, for consistency, we stick with $\partial_{\mathbf{x}} l_\gamma(\boldsymbol{w}; (\mathbf{x}, \mathrm{y})) = \{\mathbf{0}\}$ when $y \cdot \langle \boldsymbol{w}, \mathbf{x} \rangle = \gamma$. Note, that the set of points in $\mathcal{X}$ satisfying this equality is a *zero* measure set. Thus, by replacing the log-likelihood loss with the margin loss and defining its gradients in this way, it is clear that RCAD would not perturb any data points lying on or beyond the margin.

$$\partial_{\mathbf{x}} l_\gamma(\boldsymbol{w}; (\mathbf{x}, \mathbf{y})) = \begin{cases} \{-y\boldsymbol{w}\} & y \cdot \langle \boldsymbol{w}, \mathbf{x} \rangle < \gamma \\ \{-\kappa \cdot y\boldsymbol{w} \mid \kappa \in [0,1]\} & y \cdot \langle \boldsymbol{w}, \mathbf{x} \rangle = \gamma \\ \{\mathbf{0}\} & \text{otherwise} \end{cases} \tag{7}$$

The reformulated RCAD objective in Equation 6 is a non-convex optimization problem with convex constraints. Thus, different solvers may yield different solutions. The solver we choose to analyze is a typical one – taking projected gradient ascent steps to maximize the entropy term, starting from an initialization $\boldsymbol{w}^{(0)}$ that satisfies the constraint in Equation 6, and one where the iterates $\boldsymbol{w}^{(t)}$ continue to do so after every projection step.

Given the projection operator $\Pi_{\mathcal{S}}$ onto the convex set $\mathcal{S}$, and learning rate $\eta > 0$, an RCAD iteration's update rule for $\boldsymbol{w}^{(t)}$ is defined as follows:

$$\tilde{\boldsymbol{w}}^{(t+1)} = \boldsymbol{w}^{(t)} + \eta \cdot \nabla_{\boldsymbol{w}^{(t)}} \mathcal{M}_{\hat{\mathcal{D}}}(\boldsymbol{w}^{(t)})$$
$$\boldsymbol{w}^{(t+1)} = \Pi_{\|\boldsymbol{w}\|_2 \leqslant 1} \tilde{\boldsymbol{w}}^{(t+1)}$$

**Note on projection.** For simplicity we shall treat the projection operation as just renormalizing $\tilde{\boldsymbol{w}}^{(t+1)}$ to have unit norm, *i.e.*, $\|\boldsymbol{w}^{(t+1)}\|_2 = 1, \forall t \geqslant 0$. This is not necessarily restrictive. In fact, the optimal solution with the lowest test error: $\boldsymbol{w}^* \triangleq [1, 0, \ldots, 0]^\top$, satisfies $\|\boldsymbol{w}^*\|_2^2 = 1$, and thus lies on the boundary of $\mathcal{W}$. Thus, we do not lose anything by renormalizing $\tilde{\boldsymbol{w}}^{(t+1)}$ to get the next iterate $\boldsymbol{w}^{(t+1)}$. With this, we are ready to state an informal outline of our proof strategy.

**Informal proof outline.** Our main approach to prove the claim in Theorem 5.3 can be broken down and outlined in the following way:

- **Lemma A.4:** First, we shall rely on typical margin based generalization bounds to show that if our training loss is low ($\leqslant \rho/2$) and training data size $n$ is large enough, then we can bound the test error of the ERM solution by some small value $\rho$ with high probability.

- **Lemma 5.2:** Next we shall use the fact that the test error of the ERM solution is less than $\rho$, to prove a lower bound on the true feature weights $w_1$. This tells us that the true feature has been sufficiently learned, but ERM still depends on the noisy dimensions since $\|\boldsymbol{w}_2\|_2 \gg 0$, and this cannot be remedied by ERM without sampling more training data.

- **Lemma 5.4:** Now, given the ERM solution as the initialization we take steps of projected gradient ascent on the reformulated RCAD objective in Equation 6. For this procedure, under some conditions on the noise terms, margin $\gamma$ and step size $\alpha$ we show that the partial derivative of the population objective $\mathcal{M}_{\mathcal{D}}(\boldsymbol{w})$ with respect $w_1$ is along $w_1$ at every step: $\langle \frac{\partial \mathcal{M}_{\mathcal{D}}(\boldsymbol{w}^{(t)})}{\partial w_1^{(t)}}, w_1^{(t)} \rangle > 0$. On the other hand, we show that the derivative with respect to $\boldsymbol{w}_2$ is sufficiently negatively correlated with $\boldsymbol{w}_2$ at every step: $\langle \nabla_{\boldsymbol{w}_2^{(t)}} \mathcal{M}_{\mathcal{D}}(\boldsymbol{w}^{(t)}), \boldsymbol{w}_2^{(t)} \rangle < -c_1 \cdot \|\boldsymbol{w}_2^{(t)}\|_2^2$. Also, we note that projecting the iterates to a unit norm ball is sufficient to satisfy the constraints in Equation 6 after every ascent step.

- **Finally,** we use the above two sufficient conditions to arrive at the following conclusion: RCAD gradient ascent iterations on $\mathcal{M}_{\mathcal{D}}(\boldsymbol{w})$ monotonically increase the norm along the true feature ($|w_1^{(t)}| \uparrow$) and monotonically decrease the norm along the irrelevant noisy features ($\|\boldsymbol{w}_2^{(t)}\|_2 \downarrow$), thereby unlearning the spuriousness. Next, we rely on some uniform convergence arguments from prior works [7] to show that the previous two points are true even if we approximate the population gradient $\nabla_{\boldsymbol{w}} \mathcal{M}_{\mathcal{D}}(\boldsymbol{w})$, with the finite sample approximation $\nabla_{\boldsymbol{w}} \mathcal{M}_{\hat{\mathcal{D}}}(\boldsymbol{w})$, yielding the high probability finite sample guarantees in Theorem 5.3.

## A.1 Technical Lemmas

In this section we shall state some technical lemmas without proof, with references to works that contain the full proof. We shall use these in the following sections when proving our lemmas in Section 5.

**Lemma A.1** (Lipschitz functions of Gaussians [63]). *Let $X_1, \ldots, X_n$ be a vector of iid Gaussian variables and $f : \mathbb{R}^n \mapsto \mathbb{R}$ be L-Lipschitz with respect to the Euclidean norm. Then the random*

*variable $f(X) - \mathbb{E}[f(X)]$ is sub-Gaussian with parameter at most L, thus:*

$$\mathbb{P}[|f(X) - \mathbb{E}[f(X)]| \geqslant t] \leqslant 2 \cdot \exp\left(-\frac{t^2}{2L^2}\right), \quad \forall t \geqslant 0.$$

**Lemma A.2** (Hoeffding bound [63]). *Let $X_1, \ldots, X_n$ be a set of $\mu_i$ centered independent sub-Gaussians, each with parameter $\sigma_i$. Then for all $t \geqslant 0$, we have*

$$\mathbb{P}\left[\frac{1}{n}\sum_{i=1}^{n}(X_i - \mu_i) \geqslant t\right] \leqslant \exp\left(-\frac{n^2 t^2}{2\sum_{i=1}^{n}\sigma_i^2}\right).$$

**Lemma A.3** (Koltchinskii and Panchenko [29]). *If we assume $\forall f : \mathcal{X} \mapsto \mathbb{R} \in \mathcal{F}$, $\sup_{\mathbf{x}\in\mathcal{X}}|f(\mathbf{x})| \leqslant C$, then with probability at least $1 - \delta$ over sample $\hat{\mathcal{D}}$ of size $n$, given $l_\gamma(\boldsymbol{w}; (\mathbf{x}, \mathbf{y}))$ as the margin loss with some $\gamma > 0$, and $\mathcal{R}_n(\mathcal{F})$ as the empirical Rademacher complexity of class $\mathcal{F}$, the following uniform generalization bound holds $\forall f \in \mathcal{F}$,*

$$\mathbb{E}_\mathcal{D}\mathbb{1}(y \cdot \langle \boldsymbol{w}, \mathbf{x} \rangle < 0) \leqslant \mathbb{E}_{\hat{\mathcal{D}}}\mathbb{1}(l_\gamma(\boldsymbol{w}; (\mathbf{x}, \mathbf{y})) > 0) + 4\frac{\mathcal{R}_n(\mathcal{F})}{\gamma} + \sqrt{\frac{\log\log(4C/\gamma)}{n}} + \sqrt{\frac{\log 1/\delta}{2n}}.$$

### A.1.1 Some useful facts about the $\mathrm{erfc}(\cdot)$ function

In this section we define the complementary error function: $\mathrm{erfc}(x)$ and look at some useful upper and lower bounds for $\mathrm{erfc}(x)$ when $x \geqslant 0$, along with the closed form expression for its derivative. If $\Phi(x) \triangleq \mathbb{P}_{z\sim\mathcal{N}(0,1)}(z \leqslant x)$ is the cumulative distribution function of a standard normal, then the complement $\Phi_c(x) = 1 - \Phi(x)$, is related to the erfc function in the following way:

$$\Phi_c(x) = \frac{1}{\sqrt{2\pi}} \cdot \int_{z\geqslant x} \exp\left(-z^2/2\right) \cdot dz$$

$$\mathrm{erfc}(x) = 2 \cdot \Phi_c(\sqrt{2}x) = \frac{2}{\sqrt{\pi}} \cdot \int_{z\geqslant x} \exp\left(-z^2\right) \cdot dz \tag{8}$$

The erfc function is continuous and smooth, with its derivative having a closed form expression, as given by Equation 9. For our calculations we would also need convenient upper and lower bounds on the erfc function itself, given by Equation 10 and Equation 11 respectively. Full proofs for these bounds can be found in Kschischang [32].

$$\frac{d}{dx}\left(\mathrm{erfc}\left(x/\sqrt{2}\right)\right) = -\sqrt{\frac{2}{\pi}} \cdot \exp\left(-x^2/2\right) \tag{9}$$

$$\mathrm{erfc}\left(\frac{x}{\sqrt{2}}\right) \leqslant \sqrt{\frac{2}{\pi}} \cdot \frac{\exp\left(-x^2/2\right)}{x}, \qquad \text{erfc upper bound when } x \geqslant 0 \tag{10}$$

$$\mathrm{erfc}\left(\frac{x}{\sqrt{2}}\right) \geqslant 2\sqrt{\frac{2}{\pi}} \cdot \frac{\exp\left(-x^2/2\right)}{(x + \sqrt{x^2 + 4})}, \qquad \text{erfc lower bound when } x \geqslant 0 \tag{11}$$

Now, we are ready to begin proving our lemmas in Section 5. We prove them in the same order as listed in the informal proof sketch provided in Section A.

### A.2 Proof of Lemma 5.2

First, we state and prove the following lemma that bounds the generalization gap of $l_2$ norm constrained linear predictors trained by minimizing a margin loss on training data.

**Lemma A.4** (generalization gap). *With probability* $1-\delta$ *over* $\hat{\mathcal{D}}$, $\forall \boldsymbol{w} \in \mathcal{W}$,

$$\mathbb{E}_{\mathcal{D}}\left[\mathbb{1}(\mathbf{y} \cdot \langle \boldsymbol{w}, \mathbf{x} \rangle < 0)\right] \leqslant \mathbb{E}_{\hat{\mathcal{D}}}\left[\mathbb{1}(\mathbf{y} \cdot \langle \boldsymbol{w}, \mathbf{x} \rangle < \gamma)\right] + \tilde{\mathcal{O}}\left(\frac{\beta + \sqrt{\log(1/\delta)\|\tilde{\Sigma}\|_{\mathrm{op}}} + \sqrt{\|\tilde{\Sigma}\|_*}}{\gamma\sqrt{n}}\right),$$

*where* $\tilde{\mathcal{O}}$ *hides* poly-log *factors in* $\beta, \gamma, \|\tilde{\Sigma}\|$.

*Proof.*

The proof is a simple application of the margin based generalization bound in Lemma A.3. For this we first need to prove the claim in Proposition A.5 which yields a high probability bound over the function $|\langle \boldsymbol{w}, \mathbf{x} \rangle| : \mathcal{X} \mapsto \mathbb{R}$ which belongs to our class $\mathcal{W}$ when $\|\boldsymbol{w}\|_2 \leqslant 1$. Then, we shall bound the empirical Rademacher complexity of the function class $\mathcal{W}$ using Proposition A.6. These two high probability bounds are used to satisfy the conditions for Lemma A.3. Finally, we use a union bound to show that Proposition A.5, Proposition A.6 along with Lemma A.3 hold true with with probability $\geqslant 1 - \delta$, to present a final generalization gap.

In the end, we arrive at a lower bound on training sample size $n$, indicating the number of samples that are sufficient for the ERM solution obtained by minimizing the margin loss over the training data to have test error at most $\rho$.

**Proposition A.5** (high probability bound over $|\langle \boldsymbol{w}, \mathbf{x} \rangle|$). *If* $\|\boldsymbol{w}\|_2 \leqslant 1$, *then with probability* $\geqslant 1 - \frac{\delta}{2}$, *we can bound* $|\langle \boldsymbol{w}, \mathbf{x} \rangle|$ *using Lemma A.1,*

$$|\langle \boldsymbol{w}, \mathbf{x} \rangle| \leqslant \sqrt{2\|\tilde{\Sigma}\|_{\mathrm{op}} \cdot \log \frac{2}{\delta}} + \sqrt{\mathrm{tr}\left(\tilde{\Sigma}\right)} + \beta$$

*Proof.*

Since $\|w\|_2 \leqslant 1$, by Cauchy-schwartz we get $|\langle \boldsymbol{w}, \mathbf{x} \rangle| \leqslant \|\boldsymbol{w}\|_2 \|\mathbf{x}\|_2 \leqslant \|\mathbf{x}\|_2$. Now we try to get a high probability bound over $\|\mathbf{x}\|_2$. Since we know that $\mathbf{x}$ follows a multivariate Gaussian distribution, specified by Equation 5, we can use triangle inequality to conclude that $\|\mathbf{x}\|_2 \leqslant \beta + \|\tilde{\Sigma}^{1/2}\mathbf{z}\|_2$ since $\mathbf{x} = y\beta \cdot [1, 0, \ldots, 0]^\top + \tilde{\Sigma}^{1/2}\mathbf{z}$, where $\mathbf{z} \sim \mathcal{N}(\mathbf{0}, \mathbf{I_{d+1}})$.

Hence, all we need to do is get a high probability bound over $\|\tilde{\Sigma}^{1/2}\mathbf{z}\|_2$ which is a function of $d + 1$ independent Gaussian variables. Thus, we can apply the concentration bound in Lemma A.1. But before that, we need to compute the Lipschitz constant for the the function $g(\mathbf{z}) = \|\tilde{\Sigma}^{1/2}\mathbf{z}\|_2$ in the eculidean norm.

$$|\|\tilde{\Sigma}^{1/2}\mathbf{z}_1\|_2 - \|\tilde{\Sigma}^{1/2}\mathbf{z}_2\|_2| \leqslant \|\tilde{\Sigma}^{1/2}(\mathbf{z}_1 - \mathbf{z}_2)\|_2 \leqslant \sqrt{\|\tilde{\Sigma}\|_{\mathrm{op}}} \tag{12}$$

Using Lipschitz constant from Equation 12 in Lemma A.1, we arrive at the following inequality which holds with probability at least $1 - \frac{\delta}{2}$.

$$|\langle \boldsymbol{w}, \mathbf{x} \rangle| \leqslant \sqrt{2\|\tilde{\Sigma}\|_{\mathrm{op}} \cdot \log \frac{2}{\delta}} + \mathbb{E}[\|\tilde{\Sigma}^{1/2}\mathbf{z}\|_2] + \beta \tag{13}$$

We shall simplify $\mathbb{E}\left[\|\tilde{\Sigma}^{1/2}\mathbf{z}\|_2^2\right]$ in the following way:

$$\mathbb{E}\left[\|\tilde{\Sigma}^{1/2}\mathbf{z}\|_2^2\right] = \mathbb{E}\left[\mathrm{tr}\left(\mathbf{z}^\top \tilde{\Sigma}\mathbf{z}\right)\right] = \mathrm{tr}\left(\tilde{\Sigma}\,\mathbb{E}\left[\mathbf{z}\mathbf{z}^\top\right]\right) = \mathrm{tr}\left(\tilde{\Sigma}\right)$$

Now, we can use Jensen: $\mathbb{E}\left[\sqrt{\|\tilde{\Sigma}^{1/2}\mathbf{z}\|_2^2}\right] \leqslant \sqrt{\mathbb{E}\left[\|\tilde{\Sigma}^{1/2}\mathbf{z}\|_2^2\right]} = \sqrt{\mathrm{tr}\left(\tilde{\Sigma}\right)}$. Applying this result into Equation 13, we finish the proof for Proposition A.5.

The next step in proving Lemma A.4 is to bound $\mathcal{R}_n(\mathcal{W})$ which is the empirical Rademacher complexity of the class of linear predictors in $d + 1$ dimensions with $l_2$ norm $\leqslant 1$. We state an adapted form of the complexity bound (Proposition A.6) for linear predictors from Kakade et al. [27] and then apply it to our specific class of linear predictors $\mathcal{W}$.

**Proposition A.6** ($\mathcal{R}_n(\mathcal{W})$ bound for linear functions [27]). *Let $\mathcal{W}$ be a convex set inducing the set of linear functions $\mathcal{F}(\mathcal{W}) \triangleq \{\langle \boldsymbol{w}, \mathbf{x} \rangle : \mathcal{X} \mapsto \mathbb{R} \mid w \in \mathcal{W}\}$ for some input space $\mathcal{X}$, bounded in norm $\|\cdot\|$ by some value $R > 0$. Now, if $\exists$ a mapping $h : \mathcal{W} \mapsto \mathbb{R}$ that is $\kappa$-strongly with respect to the dual norm $\|\cdot\|_*$ and some subset $\mathcal{W}' \subseteq \mathcal{W}$ takes bounded values of $h(\cdot)$ i.e., $\{h(\boldsymbol{w}) \leqslant K \mid \boldsymbol{w} \in \mathcal{W}'\}$ for some $K > 0$, then the empirical Rademacher complexity of the subset $\mathcal{W}'$ given by $\mathcal{R}_n(\mathcal{F}(\mathcal{W}')) \leqslant R\sqrt{\frac{2K}{\kappa n}}$.*

Let $\|\cdot\|_2^2$ be the function $h : \mathcal{W} \mapsto \mathbb{R}$ in Proposition A.6, and we know that $\|\cdot\|_2^2$ is 2-strongly convex in $l_2$ norm. Further, take the standard $l_2$ norm as the norm over $\mathcal{X}$. So, the dual norm $\|\cdot\|_*$ is also given by $l_2$ norm. Thus, $\kappa = 2$. We also know that $\mathcal{W}$ is bounded in $\|\cdot\|_2$ by 1, based on our setup definition. Thus, $R = 1$.

Now, we look at $K$. While proving Proposition A.5 we proved a high probability bound over $\|\boldsymbol{x}\|_2$, which we shall plug into the value of $K$. Thus, from our choice of $h$ above, and the result in Proposition A.5 we can conclude that with probability $\geqslant 1 - \frac{\delta}{2}$ :

$$\mathcal{R}_n(\mathcal{W}) \leqslant \frac{\sqrt{2\|\tilde{\Sigma}\|_{\mathrm{op}} \log(2/\delta)} + \sqrt{\mathrm{tr}\left(\tilde{\Sigma}\right)} + \beta}{\sqrt{n}} \tag{14}$$

We are now ready to plugin the bounds on $\mathcal{R}_n(\mathcal{W})$ and $|\langle \boldsymbol{w}, \mathbf{x} \rangle|$ into the margin based generalization bound in Lemma A.3, where $C$ takes the value of the bound on $|\langle \boldsymbol{w}, \mathbf{x} \rangle|$. We set the sample size $n$ large enough for Lemma A.3 to hold with probability $\geqslant 1 - \frac{\delta}{2}$. Thus, by union bound the following is satisfied with probability $\geqslant 1 - \delta$.

$$\mathbb{E}_{\mathcal{D}} \mathbb{1}(y \cdot \langle \boldsymbol{w}, \mathbf{x} \rangle < 0) \leqslant \mathbb{E}_{\hat{\mathcal{D}}} \mathbb{1}(l_\gamma(\boldsymbol{w}; (\mathbf{x}, \mathbf{y})) > 0) + \tilde{\mathcal{O}}\left(\frac{\beta + \sqrt{\|\tilde{\Sigma}\|_{\mathrm{op}} \log(1/\delta)} + \sqrt{\mathrm{tr}\left(\tilde{\Sigma}\right)}}{\gamma \sqrt{n}}\right) \tag{15}$$

Corollary A.7 immediately gives us a sufficient condition on $n$ for the test error of the ERM solution to be upper bounded by $\rho$. Note, that from our constraint in Equation 6 we know that the training error of the ERM solution is at most $\rho/2$.

**Corollary A.7** (sufficient condition on $n$ for ERM test error $\leqslant \rho$). *If $n \gtrsim \frac{\beta^2 + \log(1/\delta) \cdot \|\tilde{\Sigma}\|_{\mathrm{op}} + \|\tilde{\Sigma}\|_*}{\gamma^2 \rho^2}$, then the test error of the ERM solution can be bounded by $\rho$, with probability $\geqslant 1 - \delta$ over $\hat{\mathcal{D}}$.*

*Proof.*

From the constraint in Equation 6 we know that the training loss on $\hat{\mathcal{D}}$ is at most $\rho/2$ i.e., $\mathbb{E}_{\hat{\mathcal{D}}}\left[\mathbb{1}(\mathbf{y} \cdot \langle \boldsymbol{w}, \mathbf{x} \rangle < 0)\right] \leqslant \rho/2$ in Lemma A.3. If we can upper bound the generalization error by $\frac{\rho}{2}$ then the result follows. The generalization error is bounded using Equation 15.

This completes our proof of Lemma A.4.

Let $\mu_+ \triangleq [\beta, 0, \ldots, 0] \in \mathbb{R}^{d+1}$ and $\mu_- \triangleq [-\beta, 0, \ldots, 0] \in \mathbb{R}^{d+1}$ and $\sigma_{\boldsymbol{w}}$ be as defined in Equation 5. If we assume that the test error for some $\boldsymbol{w}$ is sufficiently upper bounded: $\mathbb{E}_{\mathcal{D}}\left[\mathbb{1}(\mathbf{y} \cdot \langle \boldsymbol{w}, \mathbf{x} \rangle < 0)\right] \leqslant \rho$, then we can show that the norm along the true feature $w_1$ is also appropriately lower bounded using the following sequence of implications:

$$\mathbb{E}_{\mathcal{D}}\left[\mathbb{1}(\mathbf{y} \cdot \langle \boldsymbol{w}, \mathbf{x} \rangle < 0)\right] \;\leqslant\; \rho$$

$$\implies \frac{1}{2} \cdot \mathbb{E}_{\mathbf{z} \sim \mathcal{N}(\mathbf{0}, \mathbf{I_{d+1}})} \mathbb{1}(\langle \boldsymbol{w}, \mu_+ + \tilde{\Sigma}^{1/2}\mathbf{z} \rangle < 0) + \frac{1}{2} \cdot \mathbb{E}_{\mathbf{z} \sim \mathcal{N}(\mathbf{0}, \mathbf{I_{d+1}})} \mathbb{1}(\langle \boldsymbol{w}, \mu_- + \tilde{\Sigma}^{1/2}\mathbf{z} \rangle > 0) \leqslant \rho$$

$$\implies \frac{1}{2} \cdot \mathbb{E}_{\mathbf{z} \sim \mathcal{N}(\mathbf{0}, \sigma_{\mathbf{w}}^2)} \mathbb{1}(\beta w_1 + \mathbf{z} < 0) + \frac{1}{2} \cdot \mathbb{E}_{\mathbf{z} \sim \mathcal{N}(\mathbf{0}, \sigma_{\mathbf{w}}^2)} \mathbb{1}(-\beta w_1 + \mathbf{z} > 0) \;\leqslant\; \rho$$

$$\implies \mathbb{P}_{\mathbf{z} \sim \mathcal{N}(\mathbf{0}, \sigma_{\mathbf{w}}^2)}(\mathbf{z} > \beta w_1) \;\leqslant\; \rho$$

$$\implies \mathbb{P}_{\mathbf{z} \sim \mathcal{N}(\mathbf{0}, \mathbf{1})}\left( \mathbf{z} > \frac{\beta}{\sigma_{\boldsymbol{w}}} \cdot w_1 \right) \;\leqslant\; \rho$$

$$\implies \frac{1}{2} \cdot \operatorname{erfc}\left( \frac{\beta}{\sqrt{2}\sigma_{\boldsymbol{w}}} \cdot w_1 \right) \;\leqslant\; \rho \qquad \text{since, } \Phi_c(t) = \frac{1}{2}\operatorname{erfc}\left(t/\sqrt{2}\right)$$

$$\implies w_1 \;\geqslant\; \frac{\operatorname{erfc}^{-1}(2\rho)}{\beta} \cdot \sqrt{2}\sigma_{\boldsymbol{w}} \;\geqslant\; \frac{\operatorname{erfc}^{-1}(2\rho) \cdot \sqrt{2\sigma_{\min}(\tilde{\Sigma})}}{\beta}$$

This completes our proof of Lemma 5.2.

From Corollary A.7 and the arguments above, we arrive at the high probability result in Lemma 5.2, i.e., if $n \gtrsim \frac{\beta^2 + \log(1/\delta) \cdot \|\tilde{\Sigma}\|_{\mathrm{op}} + \|\tilde{\Sigma}\|_*}{\gamma^2 \rho^2}$, then with probability $\geqslant 1 - \delta$ over $\mathcal{D}'$, $w_1 \geqslant \frac{\operatorname{erfc}^{-1}(2\rho) \cdot \sqrt{2\sigma_{\min}(\tilde{\Sigma})}}{\beta}$.

**Corollary A.8** (ERM solution can have non-negligible dependence on $\boldsymbol{w}_2$). *For the ERM solution $\boldsymbol{w}$, in the worst cast $w_1 = \frac{\operatorname{erfc}^{-1}(2\rho) \cdot \sqrt{2\sigma_{\min}(\tilde{\Sigma})}}{\beta} < 1$ and given $\|\boldsymbol{w}\|_2 = 1$, it is easy to see that $\|\boldsymbol{w}_2\|_2 \gg 0$.*

*Proof.* Assume $\frac{\operatorname{erfc}^{-1}(2\rho) \cdot \sqrt{2\sigma_{\min}(\tilde{\Sigma})}}{\beta} \geqslant 1$. $\implies 2\rho \leqslant \operatorname{erfc}\left(\beta / \sqrt{2\sigma_{\min}(\tilde{\Sigma})}\right)$. This further implies $\rho \leqslant (1/2) \cdot \operatorname{erfc}\left(\beta/\sqrt{2\sigma_1}\right)$ which is not possible since the optimal test error achieved by $\boldsymbol{w}^*$ is $(1/2) \cdot \operatorname{erfc}\left(\beta/\sqrt{2\sigma_1}\right)$ and $\boldsymbol{w} \neq \boldsymbol{w}^*$.

### A.3 Proof of Lemma 5.4

Before proving Lemma 5.4 we derive the closed form expression for the population objective $\mathcal{M}_{\mathcal{D}}(\boldsymbol{w})$ in Lemma A.9 which we shall use repeatedly in the sections that follow.

**Lemma A.9** ($\mathcal{M}_{\mathcal{D}}(\boldsymbol{w})$ closed form). *If $\alpha \geqslant \gamma$, $\|\boldsymbol{w}\|_2 = 1$ and if we define*

$$\sigma_{\boldsymbol{w}} \triangleq \sqrt{\boldsymbol{w}^\top \tilde{\Sigma} \boldsymbol{w}}, \quad a_{\boldsymbol{w}} \triangleq \frac{\beta w_1 - \gamma}{\sigma_{\boldsymbol{w}}},$$

*then we can write the closed form for $\mathcal{M}_{\mathcal{D}}(\boldsymbol{w})$ as:*

$$\begin{aligned}
\mathcal{M}_{\mathcal{D}}(\boldsymbol{w}) \;=\; & \frac{1}{2} \cdot \exp\left( \beta w_1 - \alpha \|\boldsymbol{w}\|_2^2 + \frac{\sigma_{\boldsymbol{w}}^2}{2} \right) \cdot \operatorname{erfc}\left( \frac{\sigma_{\boldsymbol{w}} + a_{\boldsymbol{w}}}{\sqrt{2}} \right) \\
& + \frac{1}{2} \cdot \exp\left( -\beta w_1 + \frac{\sigma_{\boldsymbol{w}}^2}{2} \right) \cdot \operatorname{erfc}\left( \frac{\sigma_{\boldsymbol{w}} - a_{\boldsymbol{w}}}{\sqrt{2}} \right).
\end{aligned} \tag{16}$$

*Proof.*

Using the definition of the subgradient $\partial_{\mathbf{x}} l_\gamma(\boldsymbol{w}; (\mathbf{x}, \mathbf{y}))$ in Equation 7, the population objective can be broken down into the following integrals, where $\mathcal{D}(\mathbf{x}, \mathbf{y})$ is the measure over the space $\mathcal{X} \times \mathcal{Y}$ defined by distribution $\mathcal{D}$.

$$\mathcal{M}_{\mathcal{D}}(\boldsymbol{w}) \triangleq \mathbb{E}_{\mathcal{D}} \, \exp(-|\langle \boldsymbol{w}, \mathbf{x} + \alpha \cdot \nabla_{\boldsymbol{x}} l_{\gamma}(\boldsymbol{w}; (\mathbf{x}, \mathbf{y}))\rangle|)$$

$$= \int_{y\langle \boldsymbol{w}, \mathbf{x}\rangle < \gamma} \exp(-|\langle \boldsymbol{w}, \mathbf{x} - \alpha y \boldsymbol{w}\rangle|) \cdot \mathcal{D}(\mathbf{x}, \mathbf{y}) \; + \; \int_{y\langle \boldsymbol{w}, \mathbf{x}\rangle \geqslant \gamma} \exp(-|\langle \boldsymbol{w}, \mathbf{x}\rangle|) \cdot \mathcal{D}(\mathbf{x}, \mathbf{y})$$

$$= \frac{1}{2} \int_{\langle \boldsymbol{w}, \mathbf{x}\rangle > -\gamma} \exp(-|\boldsymbol{w}^{\top} \mathbf{x} + \alpha \, \|\boldsymbol{w}\|_2^2 \,|) \cdot \mathcal{D}(\mathbf{x} \mid \mathbf{y} = -1) \quad \left.\vphantom{\int}\right\} \textcircled{1}$$

$$+ \quad \frac{1}{2} \int_{\langle \boldsymbol{w}, \mathbf{x}\rangle < \gamma} \exp(-|\boldsymbol{w}^{\top} \mathbf{x} - \alpha \, \|\boldsymbol{w}\|_2^2 \,|) \cdot \mathcal{D}(\mathbf{x} \mid \mathbf{y} = 1) \quad \left.\vphantom{\int}\right\} \textcircled{2}$$

$$+ \quad \frac{1}{2} \int_{\langle \boldsymbol{w}, \mathbf{x}\rangle \leqslant -\gamma} \exp(-|\boldsymbol{w}^{\top} \mathbf{x}|) \cdot \mathcal{D}(\mathbf{x} \mid \mathbf{y} = -1) \quad \left.\vphantom{\int}\right\} \textcircled{3}$$

$$+ \quad \frac{1}{2} \int_{\langle \boldsymbol{w}, \mathbf{x}\rangle \geqslant \gamma} \exp(-|\boldsymbol{w}^{\top} \mathbf{x}|) \cdot \mathcal{D}(\mathbf{x} \mid \mathbf{y} = 1) \quad \left.\vphantom{\int}\right\} \textcircled{4}$$

where the final equation uses the fact that $y \sim \text{Unif}\{-1, 1\}$. We are also given that the measure $\mathcal{D}(\mathbf{x}_1 \mid \mathbf{y} = -1)$ for the conditional distribution over $\mathbf{x}_1$ given $\mathbf{y} = -1$ has distribution $\mathcal{N}(-\beta, \sigma_1^2)$. Similarly, $\mathcal{D}(\mathbf{x}_1 \mid \mathbf{y} = +1)$ follows $\mathcal{N}(\beta, \sigma_1^2)$ and measure $\mathcal{D}(\mathbf{x}_2)$ follows $\mathcal{N}(\mathbf{0}, \boldsymbol{\Sigma})$.

Thus,

$$\langle \boldsymbol{w}, \mathbf{x}\rangle \mid (\mathbf{y} = -1) \sim \mathcal{N}(-\beta w_1, \sigma_{\boldsymbol{w}}^2)$$
$$\langle \boldsymbol{w}, \mathbf{x}\rangle \mid (\mathbf{y} = 1) \sim \mathcal{N}(\beta w_1, \sigma_{\boldsymbol{w}}^2).$$

We shall substitute the above into $\textcircled{1}$, $\textcircled{2}$, $\textcircled{3}$, $\textcircled{4}$, to get:

$$\textcircled{1} = \frac{1}{2} \int_{-\beta w_1 + \sigma_{\boldsymbol{w}} z > -\gamma} \exp(-|-\beta w_1 + \sigma_{\boldsymbol{w}} z + \alpha \, \|\boldsymbol{w}\|_2^2 \,|) \, p(z) \; dz$$

$$\textcircled{2} = \frac{1}{2} \int_{\beta w_1 + \sigma_{\boldsymbol{w}} z < \gamma} \exp(-|\beta w_1 + \sigma_{\boldsymbol{w}} z - \alpha \, \|\boldsymbol{w}\|_2^2 \,|) \, p(z) \; dz$$

$$\textcircled{3} = \frac{1}{2} \int_{-\beta w_1 + \sigma_{\boldsymbol{w}} z \leqslant -\gamma} \exp(-|-\beta w_1 + \sigma_{\boldsymbol{w}} z|) \, p(z) \; dz$$

$$\textcircled{4} = \frac{1}{2} \int_{\beta w_1 + \sigma_{\boldsymbol{w}} z \geqslant \gamma} \exp(-|\beta w_1 + \sigma_{\boldsymbol{w}} z|) \, p(z) \; dz$$

where $p(z) \triangleq \frac{1}{\sqrt{2\pi}} \exp\left(-z^2/2\right)$ is the density of a standard Gaussian random variable. Now we shall look at the region where each of the four integrals are defined and apply the inequality $\alpha \geqslant \gamma$, which is stated as a condition on the step size $\alpha$ in Lemma A.9. Recall, the definition of $a_{\boldsymbol{w}} = \frac{\beta w_1 - \gamma}{\sigma_{\boldsymbol{w}}}$ and the condition $\|\boldsymbol{w}\|_2 = 1$.

Region for $\textcircled{1}$ : $(\alpha \geqslant \gamma)$ and $(-\beta w_1 + \sigma_{\boldsymbol{w}} z > -\gamma) \implies -\beta w_1 + \sigma_{\boldsymbol{w}} z + \alpha \|\boldsymbol{w}\|_2^2 > 0$

Region for $\textcircled{2}$ : $(\alpha \geqslant \gamma)$ and $(\beta w_1 + \sigma_{\boldsymbol{w}} z < \gamma) \implies \beta w_1 + \sigma_{\boldsymbol{w}} z - \alpha \|\boldsymbol{w}\|_2^2 < 0$

Region for $\textcircled{3}$ : $(\alpha \geqslant \gamma)$ and $(-\beta w_1 + \sigma_{\boldsymbol{w}} z \leqslant -\gamma) \implies -\beta w_1 + \sigma_{\boldsymbol{w}} z \leqslant 0$

Region for $\textcircled{4}$ : $(\alpha \geqslant \gamma)$ and $(\beta w_1 + \sigma_{\boldsymbol{w}} z \geqslant \gamma) \implies \beta w_1 + \sigma_{\boldsymbol{w}} z \geqslant 0$

As a result, we can further simplify the expressions into:

$$\textcircled{1} = \frac{1}{2} \int_{-\beta w_1 + \sigma_{\boldsymbol{w}} z > -\gamma} \exp(\beta w_1 - \sigma_{\boldsymbol{w}} z - \alpha \|\boldsymbol{w}\|_2^2 |) \, p(z) \; dz$$

$$\textcircled{2} = \frac{1}{2} \int_{\beta w_1 + \sigma_{\boldsymbol{w}} z < \gamma} \exp(\beta w_1 + \sigma_{\boldsymbol{w}} z - \alpha \|\boldsymbol{w}\|_2^2) \, p(z) \; dz$$

$$\textcircled{3} = \frac{1}{2} \int_{-\beta w_1 + \sigma_{\boldsymbol{w}} z \leqslant -\gamma} \exp(-\beta w_1 + \sigma_{\boldsymbol{w}} z) \, p(z) \; dz$$

$$\textcircled{4} = \frac{1}{2} \int_{\beta w_1 + \sigma_{\boldsymbol{w}} z \geqslant \gamma} \exp(-\beta w_1 - \sigma_{\boldsymbol{w}} z) \, p(z) \; dz$$

Now we shall note that $\textcircled{1} = \textcircled{2}$ and $\textcircled{3} = \textcircled{4}$, since $z$ and $-z$ are random variables that have the same probability distribution $\mathcal{N}(0,1)$. Using the definition of $\mathrm{erfc}\,(x)$ in Equation 8, we get:

$$\mathcal{M}_{\mathcal{D}}(\boldsymbol{w}) = \frac{1}{2} \left[ \exp\left( \beta w_1 - \alpha \|\boldsymbol{w}\|_2^2 + \frac{\sigma_{\boldsymbol{w}}^2}{2} \right) \cdot \mathrm{erfc}\left( \frac{\sigma_{\boldsymbol{w}} + a_{\boldsymbol{w}}}{\sqrt{2}} \right) + \right.$$
$$\left. \exp\left( -\beta w_1 + \frac{\sigma_{\boldsymbol{w}}^2}{2} \right) \cdot \mathrm{erfc}\left( \frac{\sigma_{\boldsymbol{w}} - a_{\boldsymbol{w}}}{\sqrt{2}} \right) \right].$$

This completes the proof of Lemma A.9. In the subsections that follow we look at the derivative of the population objective $\mathcal{M}_{\mathcal{D}}(\boldsymbol{w})$ with respect to weights $\boldsymbol{w}$ at each time step $t$ of RCAD's gradient ascent iterations, specifically how well the direction of the derivative is aligned/misaligned with $\boldsymbol{w}$.

### A.3.1   Deriving and bounding $\langle \frac{\partial \mathcal{M}_{\mathcal{D}}(\boldsymbol{w})}{\partial w_1}, w_1 \rangle$.

Using the reformulated expression for $\mathcal{M}_{\mathcal{D}}(\boldsymbol{w})$ derived in Lemma A.9, and given that $\|\boldsymbol{w}^{(t)}\|_2^2 = 1$ we derive the expression for $\frac{\partial \mathcal{M}_{\mathcal{D}}(\boldsymbol{w}^{(t)})}{\partial w_1^{(t)}}$ in Equation 18. Whenever convenient we drop the dependence on the superscript $(t)$. The following derivation is a simple application of the chain rule where we used the following:

$$\frac{\partial a_{\boldsymbol{w}}}{\partial w_1} = \frac{\beta + (\gamma - \beta w_1) \frac{w_1 \sigma_1^2}{\sigma_{\boldsymbol{w}}^2}}{\sigma_{\boldsymbol{w}}}, \qquad \frac{\partial \sigma_{\boldsymbol{w}}}{\partial w_1} = \frac{w_1 \sigma_1^2}{\sigma_{\boldsymbol{w}}}.$$

$$\begin{aligned}
\frac{\partial \mathcal{M}_{\mathcal{D}}(\boldsymbol{w})}{\partial w_1} = {} & \frac{1}{2} \exp\left( \beta w_1 + \frac{\sigma_{\boldsymbol{w}}^2}{2} - \alpha \right) \cdot \left( (\beta - 2\alpha w_1) \cdot \mathrm{erfc}\left( (\sigma_{\boldsymbol{w}} + a_{\boldsymbol{w}})/\sqrt{2} \right) \right. \\
& + \frac{\beta + (\gamma - \beta w_1) \frac{w_1 \sigma_1^2}{\sigma_{\boldsymbol{w}}^2}}{\sigma_{\boldsymbol{w}}} \cdot \frac{\partial \left( \mathrm{erfc}\left( (\sigma_{\boldsymbol{w}} + a_{\boldsymbol{w}})/\sqrt{2} \right) \right)}{\partial a_{\boldsymbol{w}}} + w_1 \sigma_1^2 \cdot \mathrm{erfc}\left( (\sigma_{\boldsymbol{w}} + a_{\boldsymbol{w}})/\sqrt{2} \right) \\
& + \left. \frac{w_1 \sigma_1^2}{\sigma_{\boldsymbol{w}}} \cdot \frac{\partial \left( \mathrm{erfc}\left( (\sigma_{\boldsymbol{w}} + a_{\boldsymbol{w}})/\sqrt{2} \right) \right)}{\partial \sigma_{\boldsymbol{w}}} \right) \\
& + \frac{1}{2} \exp\left( -\beta w_1 + \frac{\sigma_{\boldsymbol{w}}^2}{2} \right) \cdot \left( -\beta \cdot \mathrm{erfc}\left( (\sigma_{\boldsymbol{w}} - a_{\boldsymbol{w}})/\sqrt{2} \right) \right. \\
& + \frac{\beta + (\gamma - \beta w_1) \frac{w_1 \sigma_1^2}{\sigma_{\boldsymbol{w}}^2}}{\sigma_{\boldsymbol{w}}} \cdot \frac{\partial \left( \mathrm{erfc}\left( (\sigma_{\boldsymbol{w}} - a_{\boldsymbol{w}})/\sqrt{2} \right) \right)}{\partial a_{\boldsymbol{w}}} + w_1 \sigma_1^2 \cdot \mathrm{erfc}\left( (\sigma_{\boldsymbol{w}} - a_{\boldsymbol{w}})/\sqrt{2} \right) \\
& + \left. \frac{w_1 \sigma_1^2}{\sigma_{\boldsymbol{w}}} \cdot \frac{\partial \left( \mathrm{erfc}\left( (\sigma_{\boldsymbol{w}} - a_{\boldsymbol{w}})/\sqrt{2} \right) \right)}{\partial \sigma_{\boldsymbol{w}}} \right)
\end{aligned} \tag{17}$$

Substituting the partial derivatives for $\mathrm{erfc}\,(\cdot)$ from Equation 9 we can rewrite the above equation:

$$\frac{\partial \mathcal{M}_\mathcal{D}(\boldsymbol{w})}{\partial w_1} = \frac{1}{2} \exp\left(\beta w_1 + \frac{\sigma_{\boldsymbol{w}}^2}{2} - \alpha\right) \cdot \left((\beta - 2\alpha w_1 + w_1 \sigma_1^2) \cdot \text{erfc}\left((\sigma_{\boldsymbol{w}} + a_{\boldsymbol{w}})/\sqrt{2}\right)\right.$$

$$- \sqrt{\frac{2}{\pi}} \cdot \frac{(\beta + (\gamma - \beta w_1)\frac{w_1 \sigma_1^2}{\sigma_{\boldsymbol{w}}^2} + w_1 \sigma_1^2)}{\sigma_{\boldsymbol{w}}} \cdot \exp\left(-\frac{(\sigma_{\boldsymbol{w}} + a_{\boldsymbol{w}})^2}{2}\right)\right)$$

$$+ \frac{1}{2} \exp\left(-\beta w_1 + \frac{\sigma_{\boldsymbol{w}}^2}{2}\right) \cdot \left((-\beta + w_1 \sigma_1^2) \cdot \text{erfc}\left((\sigma_{\boldsymbol{w}} - a_{\boldsymbol{w}})/\sqrt{2}\right)\right.$$

$$+ \sqrt{\frac{2}{\pi}} \cdot \frac{(\beta + (\gamma - \beta w_1)\frac{w_1 \sigma_1^2}{\sigma_{\boldsymbol{w}}^2} - w_1 \sigma_1^2)}{\sigma_{\boldsymbol{w}}} \cdot \exp\left(-\frac{(\sigma_{\boldsymbol{w}} - a_{\boldsymbol{w}})^2}{2}\right)\right) \tag{18}$$

Now we shall bound the term $\langle w_1, \frac{\partial \mathcal{M}_\mathcal{D}(\boldsymbol{w})}{\partial w_1}\rangle$ and show that $\langle w_1, \frac{\partial \mathcal{M}_\mathcal{D}(\boldsymbol{w})}{\partial w_1}\rangle > 0$ with high probability over the dataset $\mathcal{D}'$.

From our assumption on the margin term $\gamma$ in Theorem 5.3 we can bound $a_{\boldsymbol{w}} \leqslant c_0$, where $c_0 > 0$,

$$\gamma + c_0 \sigma_{\min}(\tilde{\Sigma}) \geqslant \beta \implies \gamma - \beta w_1 \geqslant -c_0 \sigma_{\boldsymbol{w}} \implies a_{\boldsymbol{w}} \leqslant c_0 \tag{19}$$

$$\implies (\beta w_1 - \gamma)\frac{w_1 \sigma_1^2}{\sigma_{\boldsymbol{w}}^2} \leqslant c_0 \cdot \frac{\sigma_1^2}{\sqrt{\sigma_{\min}(\tilde{\Sigma})}}$$

Thus, when $\beta$ is large enough such that $\beta \gtrsim \sigma_1^2$, we get $(\beta + (\gamma - \beta w_1)\frac{w_1 \sigma_1^2}{\sigma_{\boldsymbol{w}}^2} - w_1 \sigma_1^2) > 0$. This is exactly the assumption regarding the separation between the class means ($\beta$) made in Theorem 5.3, i.e., we are given that $\beta \gtrsim \|\tilde{\Sigma}\|_{\text{op}} \implies \beta \gtrsim \sigma_1^2$. Finally, we know that $w_1 > 0$ from our initialization condition in Lemma 5.2. Thus, when showing $\langle w_1, \frac{\partial \mathcal{M}_\mathcal{D}(\boldsymbol{w})}{\partial w_1}\rangle > 0$, we can ignore corresponding terms in Equation 18, and it is sufficient to show:

$$\exp(\beta w_1 - \alpha) \cdot \left((\beta w_1 - 2\alpha w_1^2 + w_1^2 \sigma_1^2) \cdot \text{erfc}\left((\sigma_{\boldsymbol{w}} + a_{\boldsymbol{w}})/\sqrt{2}\right)\right.$$

$$- \sqrt{\frac{2}{\pi}} \cdot \frac{(\beta w_1 + (\gamma - \beta w_1)\frac{w_1^2 \sigma_1^2}{\sigma_{\boldsymbol{w}}^2} + w_1^2 \sigma_1^2)}{\sigma_{\boldsymbol{w}}} \cdot \exp\left(-\frac{(\sigma_{\boldsymbol{w}} + a_{\boldsymbol{w}})^2}{2}\right)\right)$$

$$+ \exp(-\beta w_1) \cdot \left((-\beta w_1 + w_1^2 \sigma_1^2) \cdot \text{erfc}\left((\sigma_{\boldsymbol{w}} - a_{\boldsymbol{w}})/\sqrt{2}\right)\right) > 0 \tag{20}$$

First, we will show that $a_{\boldsymbol{w}} + \sigma_{\boldsymbol{w}} \geqslant 0$. Then, we will prove the critical condition in Equation 21. This would imply that the term multiplied with $\exp(\beta w_1 - \alpha)$ is positive, and since for large enough $\beta$, we get $\exp(\beta w_1 - \alpha) \gg \exp(-\beta w_1)$, we can recover the inequality in Equation 20.

$$(\beta w_1 - 2\alpha w_1^2 + w_1^2 \sigma_1^2) \cdot \text{erfc}\left((\sigma_{\boldsymbol{w}} + a_{\boldsymbol{w}})/\sqrt{2}\right)$$

$$- \sqrt{\frac{2}{\pi}} \cdot \frac{(\beta w_1 + (\gamma - \beta w_1)\frac{w_1^2 \sigma_1^2}{\sigma_{\boldsymbol{w}}^2} + w_1^2 \sigma_1^2)}{\sigma_{\boldsymbol{w}}} \cdot \exp\left(-\frac{(\sigma_{\boldsymbol{w}} + a_{\boldsymbol{w}})^2}{2}\right) > 0 \tag{21}$$

From Theorem 5.3, we are given that $n \gtrsim \frac{\beta^2 + \log(1/\delta) \cdot \|\tilde{\Sigma}\|_{\text{op}} + \|\tilde{\Sigma}\|_*}{\gamma^2 \text{erfc}\left(K\alpha/\sqrt{2\sigma_{\min}(\tilde{\Sigma})}\right)^2}$. Applying Corollary A.7 we get that $\beta w_1 \geqslant K\alpha$ at initialization using the following sequence of arguments:

$$n \gtrsim \frac{\beta^2 + \log(1/\delta) \cdot \|\tilde{\Sigma}\|_{\mathrm{op}} + \|\tilde{\Sigma}\|_*}{\gamma^2 \mathrm{erfc}\left(K\alpha/\sqrt{2\sigma_{\min}(\tilde{\Sigma})}\right)^2} \implies \rho \leqslant \mathrm{erfc}\left(\frac{K\alpha}{\sqrt{2\sigma_{\min}(\tilde{\Sigma})}}\right)$$

$$\implies \mathrm{erfc}^{-1}(\rho)\sqrt{2\sigma_{\min}(\tilde{\Sigma})} \geqslant K\alpha \implies \beta w_1 \geqslant K\alpha \tag{22}$$

Now, $a_{\boldsymbol{w}} + \sigma_{\boldsymbol{w}} = \frac{\beta w_1 - \gamma}{\sigma_{\boldsymbol{w}}} + \sigma_{\boldsymbol{w}}$. Since, $\beta w_1 \geqslant K\alpha$ for a sufficiently large $K$ that we can choose, and given that the margin $\gamma \leqslant \alpha$, it is easy to see that $a_{\boldsymbol{w}} + \sigma_{\boldsymbol{w}} > 0$. Thus, we can substitute $\mathrm{erfc}\left((\sigma_{\boldsymbol{w}} + a_{\boldsymbol{w}})/\sqrt{2}\right)$ with its lower bound from Equation 11 into Equation 21 to get:

$$(\beta w_1 - 2\alpha w_1^2 + w_1^2 \sigma_1^2) \cdot 2\sqrt{\frac{2}{\pi}} \cdot \frac{\exp\left(-(a_{\boldsymbol{w}} + \sigma_{\boldsymbol{w}})^2/2\right)}{a_{\boldsymbol{w}} + \sigma_{\boldsymbol{w}} + \sqrt{(a_{\boldsymbol{w}} + \sigma_{\boldsymbol{w}})^2 + 4}}$$

$$-\sqrt{\frac{2}{\pi}} \cdot \frac{(\beta w_1 + (\gamma - \beta w_1)\frac{w_1^2 \sigma_1^2}{\sigma_{\boldsymbol{w}}^2} + w_1^2 \sigma_1^2)}{\sigma_{\boldsymbol{w}}} \cdot \exp\left(-\frac{(\sigma_{\boldsymbol{w}} + a_{\boldsymbol{w}})^2}{2}\right) \tag{23}$$

Simplifying the above expression by taking out common positive terms, we conclude that it is sufficient to prove the following lower bound, to effectively prove Equation 21.

$$\frac{2(\beta w_1 - 2\alpha w_1^2 + w_1^2 \sigma_1^2)}{a_{\boldsymbol{w}} + \sigma_{\boldsymbol{w}} + \sqrt{(a_{\boldsymbol{w}} + \sigma_{\boldsymbol{w}})^2 + 4}} - \frac{(\beta w_1 + (\gamma - \beta w_1)\frac{w_1^2 \sigma_1^2}{\sigma_{\boldsymbol{w}}^2} + w_1^2 \sigma_1^2)}{\sigma_{\boldsymbol{w}}}$$

Recall the margin assumption in Theorem 5.3, and the corresponding implication in Equation 19. Thus, we know that $a_{\boldsymbol{w}} \leqslant c_0$, since $a_{\boldsymbol{w}} \leqslant \frac{\beta - \gamma}{\sqrt{\sigma_{\min}(\tilde{\Sigma})}}$. Thus, $\frac{a_{\boldsymbol{w}} + \sigma_{\boldsymbol{w}} + \sqrt{(a_{\boldsymbol{w}} + \sigma_{\boldsymbol{w}})^2 + 4}}{2\sigma_{\boldsymbol{w}}} \leqslant \tilde{c} \triangleq \frac{c_0 + \sqrt{\|\tilde{\Sigma}\|_{\mathrm{op}}} + \sqrt{(c_0 + \sqrt{\|\tilde{\Sigma}\|_{\mathrm{op}}})^2 + 4}}{2\sqrt{\sigma_{\min}(\tilde{\Sigma})}}$. Let $\beta w_1 \geqslant K\alpha \geqslant \mathrm{erfc}^{-1}(2\rho) \cdot \sqrt{2\sigma_{\min}(\tilde{\Sigma})}$, which is true with high probability at initialization, as we derived in Equation 22. Thus for some arbitrary constant $10/9$ that need only be slightly greater than 1, if

$$\gamma < \mathrm{erfc}^{-1}(2\rho) \cdot \sqrt{2\sigma_{\min}(\tilde{\Sigma})} - \beta^2 \left(1 - \frac{1}{(10/9)\tilde{c}}\right) \cdot \frac{\|\tilde{\Sigma}\|_{\mathrm{op}}}{\sigma_1^2 \left(\mathrm{erfc}^{-1}(2\rho) \cdot \sqrt{2\sigma_{\min}(\tilde{\Sigma})}\right)} \tag{24}$$

$$\implies \gamma < \beta w_1 - \beta^2 \left(1 - \frac{1}{(10/9)\tilde{c}}\right) \cdot \frac{\|\tilde{\Sigma}\|_{\mathrm{op}}}{\sigma_1^2 \left(\mathrm{erfc}^{-1}(2\rho) \cdot \sqrt{2\sigma_{\min}(\tilde{\Sigma})}\right)}$$

$$\implies \gamma < \beta w_1 - \beta \left(1 - \frac{1}{(10/9)\tilde{c}}\right) \cdot \frac{\|\tilde{\Sigma}\|_{\mathrm{op}}}{\sigma_1^2 w_1} < \beta w_1 - \beta \left(1 - \frac{1}{(10/9)\tilde{c}}\right) \cdot \frac{\sigma_{\boldsymbol{w}}^2}{w_1 \sigma_1^2}$$

$$\implies \frac{\beta w_1 - \gamma}{\sigma_{\boldsymbol{w}}} \cdot \frac{w_1 \sigma_1^2}{\sigma_{\boldsymbol{w}}} > \beta \left(1 - \frac{1}{(10/9)\tilde{c}}\right)$$

$$\implies w_1 \cdot \frac{\beta w_1 - \gamma}{\sigma_{\boldsymbol{w}}} \cdot \frac{w_1 \sigma_1^2}{\sigma_{\boldsymbol{w}}} > w_1 \cdot \beta \left(1 - \frac{1}{(10/9)\tilde{c}}\right)$$

$$\implies \beta w_1 - a_{\boldsymbol{w}} \frac{\sigma_1^2 w_1^2}{\sigma_{\boldsymbol{w}}} < \frac{9}{10} \frac{\beta w_1}{\tilde{c}} \tag{25}$$

Using the definition of $\tilde{c}$ and Equation 25, we can derive the following:

$$\left( \frac{2\beta w_1}{a_{\boldsymbol{w}} + \sigma_{\boldsymbol{w}} + \sqrt{(a_{\boldsymbol{w}} + \sigma_{\boldsymbol{w}})^2 + 4}} - \frac{(\beta w_1 + (\gamma - \beta w_1)\frac{w_1^2 \sigma_1^2}{\sigma_{\boldsymbol{w}}^2})}{\sigma_{\boldsymbol{w}}} \right)$$

$$\geqslant \frac{2}{a_{\boldsymbol{w}} + \sigma_{\boldsymbol{w}} + \sqrt{(a_{\boldsymbol{w}} + \sigma_{\boldsymbol{w}})^2 + 4}} \cdot \left( \beta w_1 - \tilde{c} \left( \beta w_1 - a_{\boldsymbol{w}} \frac{w_1^2 \sigma_1^2}{\sigma_{\boldsymbol{w}}} \right) \right)$$

$$\geqslant \frac{2}{a_{\boldsymbol{w}} + \sigma_{\boldsymbol{w}} + \sqrt{(a_{\boldsymbol{w}} + \sigma_{\boldsymbol{w}})^2 + 4}} \cdot \left( \beta w_1 - \tilde{c}\frac{9}{10}\frac{\beta w_1}{\tilde{c}} \right) \geqslant c_3 \cdot \beta w_1 \qquad (26)$$

for some constant $c_3 > 0$.

Now, let us come back to Equation 23. Under some conditions on $\gamma$ in Equation 24, we arrived at the final result in Equation 26. This, further implies that Equation 23 is strictly greater than zero, when $\beta w_1 \geqslant K\alpha$ and $\beta \gtrsim \|\tilde{\Sigma}\|_{\text{op}}$, where the former is proven and the latter is assumed. This finally recovers the critical condition in Equation 21.

Before, we look at proving the final bound in Equation 20, let us quickly revisit the condition on $\gamma$ in Equation 24 that is sufficient to make the claims above. If we allow $n \gtrsim \frac{\beta^2 + \log(1/\delta) \cdot \|\tilde{\Sigma}\|_{\text{op}} + \|\tilde{\Sigma}\|_*}{\gamma^2 \rho^2}$, then we can allow $\rho$ to be as small as we want for the right hand side in Equation 24 to be positive. In addition, we can also control the term $\|\tilde{\Sigma}\|_{\text{op}}$ by assuming that it is sufficiently bounded. Thus, it is not hard to see that for any large enough $\beta$, if the training data is large enough and/or noise is bounded, then there exists a valid value of the margin $\gamma$ satisfying Equation 24. Hence, in a way this condition on the margin is quite benign compared to the assumption in Theorem 5.3, which yielded the upper bound over $a_{\boldsymbol{w}}$ as a consequence of Equation 19.

Given that the bound in Equation 21 is satisfied, we can easily verify that Equation 20 would also be true since $\beta w_1 \geqslant K\alpha$ and $\beta \gtrsim \|\tilde{\Sigma}\|_{\text{op}}$. Thus, $\exists K_0$ that is large enough so that for any constant $c''' > 0$ and $\alpha > 0$:

$$\exp\left((K-1)\alpha\right) \cdot c''' - K\alpha \cdot \exp\left(-K\alpha\right) > 0, \quad \forall K \geqslant K_0$$

This completes the proof for the first part of Lemma 5.4, i.e., we have shown $\langle w_1, \frac{\partial \mathcal{M}_{\mathcal{D}}(\boldsymbol{w})}{\partial w_1} \rangle > 0$ at initialization, and for any $w_1^{(t)} \geqslant w_1^{(0)}$. We will see in the proof of Theorem 5.3 that this inequality is also true, since $|w_1|$ increases monotonically whenever $\langle w_1, \frac{\partial \mathcal{M}_{\mathcal{D}}(\boldsymbol{w})}{\partial w_1} \rangle > 0$. Thus, by an induction argument $\langle w_1^{(t)}, \frac{\partial \mathcal{M}_{\mathcal{D}}(\boldsymbol{w}^{(t)})}{\partial w_1^{(t)}} \rangle > 0, \ \forall t \geqslant 0$. In the following section we look at the second part of Lemma 5.4, involving $\nabla_{\boldsymbol{w}_2} \mathcal{M}_{\mathcal{D}}(\boldsymbol{w})$.

### A.3.2 Deriving and bounding $\langle \nabla_{\boldsymbol{w}_2} \mathcal{M}_{\mathcal{D}}(\boldsymbol{w}), \boldsymbol{w}_2 \rangle$.

Using the reformulated expression for $\mathcal{M}_{\mathcal{D}}(\boldsymbol{w})$ derived in Lemma A.9, and given that $\|\boldsymbol{w}^{(t)}\|_2^2 = 1$ we derive the expression for $\nabla_{\boldsymbol{w}_2} \mathcal{M}_{\mathcal{D}}(\boldsymbol{w})$ in Equation 28. The following derivation is a simple application of the chain rule where we used the following:

$$\nabla_{\boldsymbol{w}_2} a_{\boldsymbol{w}} = \frac{(\gamma - \beta w_1)}{\sigma_{\boldsymbol{w}}} \cdot \frac{\Sigma \boldsymbol{w}_2}{\sigma_{\boldsymbol{w}}^2}, \qquad \nabla_{\boldsymbol{w}_2} \sigma_{\boldsymbol{w}} = \frac{\Sigma \boldsymbol{w}_2}{\sigma_{\boldsymbol{w}}}.$$

$$\nabla_{\boldsymbol{w_2}}\mathcal{M}_{\mathcal{D}}(\boldsymbol{w}) = \frac{1}{2}\exp\left(\beta w_1 + \frac{\sigma_{\boldsymbol{w}}^2}{2} - \alpha\right) \cdot \left(\mathrm{erfc}\left((\sigma_{\boldsymbol{w}} + a_{\boldsymbol{w}})/\sqrt{2}\right) \cdot (\Sigma - 2\alpha\mathbf{I_d})\mathbf{w_2}\right.$$

$$+ \frac{\partial\left(\mathrm{erfc}\left((\sigma_{\boldsymbol{w}} + a_{\boldsymbol{w}})/\sqrt{2}\right)\right)}{\partial\sigma_{\boldsymbol{w}}} \cdot \frac{\Sigma\boldsymbol{w_2}}{\sigma_{\boldsymbol{w}}}$$

$$+ \frac{\partial\left(\mathrm{erfc}\left((\sigma_{\boldsymbol{w}} + a_{\boldsymbol{w}})/\sqrt{2}\right)\right)}{\partial a_{\boldsymbol{w}}} \cdot \frac{(\gamma - \beta w_1)}{\sigma_{\boldsymbol{w}}^2}\frac{\Sigma\boldsymbol{w_2}}{\sigma_{\boldsymbol{w}}}\right)$$

$$+ \frac{1}{2}\exp\left(-\beta w_1 + \frac{\sigma_{\boldsymbol{w}}^2}{2}\right) \cdot \left(\mathrm{erfc}\left((\sigma_{\boldsymbol{w}} - a_{\boldsymbol{w}})/\sqrt{2}\right)\right.$$

$$+ \frac{1}{\sigma_{\boldsymbol{w}}} \cdot \frac{\partial\left(\mathrm{erfc}\left((\sigma_{\boldsymbol{w}} - a_{\boldsymbol{w}})/\sqrt{2}\right)\right)}{\partial\sigma_{\boldsymbol{w}}}$$

$$\left. - \frac{a_{\boldsymbol{w}}}{\sigma_{\boldsymbol{w}}^2} \cdot \frac{\partial\left(\mathrm{erfc}\left((\sigma_{\boldsymbol{w}} - a_{\boldsymbol{w}})/\sqrt{2}\right)\right)}{\partial a_{\boldsymbol{w}}}\right) \cdot \Sigma\boldsymbol{w_2} \tag{27}$$

Substituting the partial derivatives for $\mathrm{erfc}(\cdot)$ from Equation 9 we can rewrite the above equation:

$$\nabla_{\boldsymbol{w_2}}\mathcal{M}_{\mathcal{D}}(\boldsymbol{w}) = \frac{1}{2}\exp\left(\beta w_1 + \frac{\sigma_{\boldsymbol{w}}^2}{2} - \alpha\right) \cdot \left(\mathrm{erfc}\left((\sigma_{\boldsymbol{w}} + a_{\boldsymbol{w}})/\sqrt{2}\right) \cdot (\Sigma - 2\alpha\mathbf{I}_d)\boldsymbol{w_2}\right.$$

$$\left. - \sqrt{\frac{2}{\pi}}\exp\left(-\frac{(\sigma_{\boldsymbol{w}} + a_{\boldsymbol{w}})^2}{2}\right) \cdot \frac{\Sigma\boldsymbol{w_2}}{\sigma_{\boldsymbol{w}}} + \sqrt{\frac{2}{\pi}}\exp\left(-\frac{(\sigma_{\boldsymbol{w}} + a_{\boldsymbol{w}})^2}{2}\right) \cdot \frac{a_{\boldsymbol{w}}\Sigma\boldsymbol{w_2}}{\sigma_{\boldsymbol{w}}^2}\right)$$

$$+ \frac{1}{2}\exp\left(-\beta w_1 + \frac{\sigma_{\boldsymbol{w}}^2}{2}\right) \cdot \left(\mathrm{erfc}\left((\sigma_{\boldsymbol{w}} - a_{\boldsymbol{w}})/\sqrt{2}\right)\right.$$

$$\left. - \frac{1}{\sigma_{\boldsymbol{w}}}\sqrt{\frac{2}{\pi}}\exp\left(-\frac{(\sigma_{\boldsymbol{w}} - a_{\boldsymbol{w}})^2}{2}\right) - \frac{a_{\boldsymbol{w}}}{\sigma_{\boldsymbol{w}}^2}\sqrt{\frac{2}{\pi}}\exp\left(-\frac{(\sigma_{\boldsymbol{w}} - a_{\boldsymbol{w}})^2}{2}\right)\right) \cdot \Sigma\boldsymbol{w_2} \tag{28}$$

**Lemma A.10** ($\|\nabla_{\boldsymbol{w_2}}\mathcal{M}_{\mathcal{D}}(\boldsymbol{w})\|_2$ is bounded). *Given the expression for $\nabla_{\boldsymbol{w_2}}\mathcal{M}_{\mathcal{D}}(\boldsymbol{w})$ in Equation 28, when $\|\boldsymbol{w}\|_2 = 1$, and the conditions on $\gamma, \alpha$ in Theorem 5.3 hold true then $\|\nabla_{\boldsymbol{w_2}}\mathcal{M}_{\mathcal{D}}(\boldsymbol{w})\|_2^2 \leqslant c_1'\|\boldsymbol{w_2}\|_2^2$ for some $c_1' > 0$.*

*Proof.*

Since $\|\boldsymbol{w}\|_2 = 1$, for any fixed value of $\beta$ and $\alpha$, it is easy to see from $Equation$ 28 that the upper bound on $\langle\nabla_{\boldsymbol{w_2}}\mathcal{M}_{\mathcal{D}}(\boldsymbol{w}), \nabla_{\boldsymbol{w_2}}\mathcal{M}_{\mathcal{D}}(\boldsymbol{w})\rangle$, will only have terms involving some positive scalar times $\|\boldsymbol{w_2}\|_2^2$, where the scalar depends on $\sigma_{\min}(\tilde{\Sigma}), \|\tilde{\Sigma}\|_{\mathrm{op}}, \beta, \alpha, \gamma$ and $c_0$. Since these are finite and positive, and erfc is bounded, the scalar is bounded above for any value taken by $\boldsymbol{w}$. From this, we can conclude that $\|\nabla_{\boldsymbol{w_2}}\mathcal{M}_{\mathcal{D}}(\boldsymbol{w})\|_2^2 \leqslant c_1'\|\boldsymbol{w_2}\|_2^2$ for some $c_1' > 0$ that is independent of $\boldsymbol{w}$.

Now, we look at $\langle\boldsymbol{w_2}, \nabla_{\boldsymbol{w_2}}\mathcal{M}_{\mathcal{D}}(\boldsymbol{w})\rangle$ and show that $\exists c_1 > 0$, such that $\langle\boldsymbol{w_2}, \nabla_{\boldsymbol{w_2}}\mathcal{M}_{\mathcal{D}}(\boldsymbol{w})\rangle < -c_1 \cdot \|\boldsymbol{w_2}\|_2^2$. Since, erfc and exp take non-negative values, and $\Sigma$ is positive semi-definite covariance matrix, we note that:

$$-\sqrt{\frac{2}{\pi}} \cdot \left(\frac{1}{2}\exp\left(\beta w_1 + \frac{\sigma_{\boldsymbol{w}}^2}{2} - \alpha\right) \cdot \frac{\boldsymbol{w_2}^\top\Sigma\boldsymbol{w_2}}{\sigma_{\boldsymbol{w}}} \cdot \exp\left(-\frac{(\sigma_{\boldsymbol{w}} + a_{\boldsymbol{w}})^2}{2}\right)\right.$$

$$\left. + \frac{1}{2}\exp\left(-\beta w_1 + \frac{\sigma_{\boldsymbol{w}}^2}{2}\right) \cdot \frac{\boldsymbol{w_2}^\top\Sigma\boldsymbol{w_2}}{\sigma_{\boldsymbol{w}}} \cdot \exp\left(-\frac{(\sigma_{\boldsymbol{w}} - a_{\boldsymbol{w}})^2}{2}\right)\right) < 0$$

Recall Equation 22 where we derived that $\beta w_1 \geqslant K$ for some large $K > 0$, and we assume $K\alpha \geqslant K\gamma$ in Theorem 5.3. Thus, $a_{\boldsymbol{w}} = \frac{\beta w_1 - \gamma}{\sigma_{\boldsymbol{w}}} > 0$, with high probability given sufficient training samples. Hence,

$$-\sqrt{\frac{2}{\pi}} \cdot \frac{1}{2}\exp\left(-\beta w_1 + \frac{\sigma_{\boldsymbol{w}}^2}{2}\right) \cdot \frac{a_{\boldsymbol{w}}}{\sigma_{\boldsymbol{w}}^2} \cdot \boldsymbol{w_2}^\top\Sigma\boldsymbol{w_2} \cdot \exp\left(-\frac{(\sigma_{\boldsymbol{w}} - a_{\boldsymbol{w}})^2}{2}\right) < 0$$

Thus, the only remaining terms are:

$$\frac{1}{2}\exp\left(\beta w_1 + \frac{\sigma_{\boldsymbol{w}}^2}{2} - \alpha\right) \cdot \left(\mathrm{erfc}\left((\sigma_{\boldsymbol{w}} + a_{\boldsymbol{w}})/\sqrt{2}\right) \cdot \boldsymbol{w}_2^\top(\Sigma - 2\alpha\mathbf{I}_d)\boldsymbol{w}_2\right.$$

$$+ \sqrt{\frac{2}{\pi}}\exp\left(-\frac{(\sigma_{\boldsymbol{w}} + a_{\boldsymbol{w}})^2}{2}\right) \cdot \frac{a_{\boldsymbol{w}}}{\sigma_{\boldsymbol{w}}^2}\boldsymbol{w}_2^\top\Sigma\boldsymbol{w}_2\left.\right)$$

$$+ \frac{1}{2}\exp\left(-\beta w_1 + \frac{\sigma_{\boldsymbol{w}}^2}{2}\right) \cdot \mathrm{erfc}\left((\sigma_{\boldsymbol{w}} - a_{\boldsymbol{w}})/\sqrt{2}\right) \cdot \boldsymbol{w}_2^\top\Sigma\boldsymbol{w}_2 \tag{29}$$

Since $a_{\boldsymbol{w}} + \sigma_{\boldsymbol{w}} > 0$, we use the upper bound from Equation 10 on $\mathrm{erfc}\left((\sigma_{\boldsymbol{w}} + a_{\boldsymbol{w}})/\sqrt{2}\right)$. Thus, we realize that if Equation 30 holds true, then we get the desired result: $\langle \boldsymbol{w}_2, \nabla_{\boldsymbol{w}_2}\mathcal{M}_\mathcal{D}(\boldsymbol{w})\rangle \leqslant -c_1\|\boldsymbol{w}_2\|_2^2$.

$$\frac{1}{2}\exp\left(\beta w_1 + \frac{\sigma_{\boldsymbol{w}}^2}{2} - \alpha\right) \cdot \sqrt{\frac{2}{\pi}} \cdot \exp\left(-\frac{(\sigma_{\boldsymbol{w}} + a_{\boldsymbol{w}})^2}{2}\right) \cdot \left(\boldsymbol{w}_2^\top\left(\Sigma\left(1 + \frac{a_{\boldsymbol{w}}}{\sigma_{\boldsymbol{w}}^2}\right) - 2\alpha\mathbf{I}_d\right)\boldsymbol{w}_2\right)$$

$$+ \frac{1}{2}\exp\left(-\beta w_1 + \frac{\sigma_{\boldsymbol{w}}^2}{2}\right) \cdot \mathrm{erfc}\left((\sigma_{\boldsymbol{w}} - a_{\boldsymbol{w}})/\sqrt{2}\right) \cdot \boldsymbol{w}_2^\top\Sigma\boldsymbol{w}_2 \leqslant -c_1\|\boldsymbol{w}_2\|_2^2 \tag{30}$$

Recall our assumption on the step size $\alpha$ in Theorem 5.3: $\alpha \gtrsim \|\tilde{\Sigma}\|_{\mathrm{op}}$. Also, from our assumption on margin $\gamma$ in Theorem 5.3, we know that $\gamma + c_0\sigma_{\min}(\tilde{\Sigma}) \geqslant \beta$, which translates to $a_{\boldsymbol{w}} \leqslant c_0$ using Equation 19. Since $\beta \gtrsim \|\tilde{\Sigma}\|_{\mathrm{op}}$, we get $\gamma + c_0\sigma_{\min}(\tilde{\Sigma}) \gtrsim \|\tilde{\Sigma}\|_{\mathrm{op}}$. When, $\sigma_{\min}(\tilde{\Sigma})$ is lower bounded by some positive constant $c_4$, then $\frac{1}{2}\left(\|\Sigma\|_{\mathrm{op}} + \frac{c_0}{\sigma_{\min}(\tilde{\Sigma})}\right) \lesssim \left(c_0\sigma_{\min}(\tilde{\Sigma}) + \frac{c_0\sigma_{\min}(\tilde{\Sigma})}{c_4^2}\right) \lesssim \|\tilde{\Sigma}\|_{\mathrm{op}}$. The margin condition on $\gamma$ trivially implies this lower bound. Still, let us discuss why this is not a restricting, rather a necessary condition.

In order for RCAD to identify and unlearn noisy dimensions, it is needed that the feature variance is not diminishing. When $\sigma_{\min}(\tilde{\Sigma}) \to 0$, then both the sampled data, and the self-generated examples by RCAD would have point mass distributions, along certain directions, which can result in some degenerate cases. Essentially we are assuming the covariance matrix $\Sigma$ to be positive definite. This is not a strong assumption and is needed to provably unlearn components along the $d$ noisy dimensions. If the variance is 0 along some direction, and if the initialization for ERM/RCAD has a non-zero component along that direction, there is no way to reduce this component to lower values with zero mean zero variance features along this direction. Thus, $\Sigma \succ \mathbf{0}$. In the case where $\sigma_1 = 0$, the term $\sigma_{\boldsymbol{w}}^2$ would be vanishingly close 0 only when $\|\boldsymbol{w}_2\| \to 0$ i.e., we are arbitrarily close to $\boldsymbol{w}^*$.

By choosing $\alpha \gtrsim \|\tilde{\Sigma}\|_{\mathrm{op}}$, we can ensure that $\left(\Sigma\left(1 + \frac{a_{\boldsymbol{w}}}{\sigma_{\boldsymbol{w}}^2}\right) - 2\alpha\mathbf{I}_d\right)$ is a negative definite matrix with all negative singular values. Thus, we can show that $\exists c'', c''' > 0$ wherein, the first term in Equation 30 scales as $-\exp\left((K-1)\alpha\right) \cdot c''\|\boldsymbol{w}_2\|_2^2$ and the second term scales as $c'''\|\boldsymbol{w}_2\|_2^2 \cdot \exp\left(-K\alpha\right)$. Now, recall Equation 22 where we derived $\beta w_1 \geqslant K\alpha$ and $\beta \gtrsim \|\tilde{\Sigma}\|_{\mathrm{op}}$. Thus, $\exists K_0$ that is large enough so that for any constants $c_1, c'', c''' > 0$ and $\alpha > 0$:

$$-\exp\left((K-1)\alpha\right) \cdot c''\|\boldsymbol{w}_2\|_2^2 + c'''\|\boldsymbol{w}_2\|_2^2 \cdot \exp\left(-K\alpha\right) \leqslant -c_1\|\boldsymbol{w}_2\|_2^2, \quad \forall K \geqslant K_0$$

Thus, we conclude that $\langle \boldsymbol{w}_2, \nabla_{\boldsymbol{w}_2}\mathcal{M}_\mathcal{D}(\boldsymbol{w})\rangle \leqslant -c_1 \cdot \|\boldsymbol{w}_2\|_2^2$, for some $c_1 > 0$. This, completes our proof for the second part of Lemma 5.4, and by making the same induction argument that we saw at the end of the previous section, we get $\langle \boldsymbol{w}_2^{(t)}, \nabla_{\boldsymbol{w}_2^{(t)}}\mathcal{M}_\mathcal{D}(\boldsymbol{w}^{(t)})\rangle \leqslant -c_1 \cdot \|\boldsymbol{w}_2^{(t)}\|_2^2, \forall t \geqslant 0$. This completes our proof of Lemma 5.4.

### A.4  Proof of Theorem 5.3

From the previous sections we concluded the following results with respect to the population objective $\mathcal{M}_\mathcal{D}(\boldsymbol{w})$: *(i)* $\langle \frac{\partial\mathcal{M}_\mathcal{D}(\boldsymbol{w}^{(t)})}{\partial w_1^{(t)}}, w_1^{(t)}\rangle > 0$; and *(ii)* $\langle \nabla_{\boldsymbol{w}_2^{(t)}}\mathcal{M}_\mathcal{D}(\boldsymbol{w}^{(t)}), \boldsymbol{w}_2^{(t)}\rangle < -c_1 \cdot \|\boldsymbol{w}_2^{(t)}\|_2^2$. Now, Lemma A.11 below tells us that there is an appropriate learning rate $\eta$ such that if we do projected

gradient ascent on the population objective $\mathcal{M}_{\mathcal{D}}(\boldsymbol{w})$, starting from the initialization satisfying Lemma 5.2, then the iterate $\boldsymbol{w}^{(T)}$ would be $\epsilon$ close to $\boldsymbol{w}^*$ in $T = \mathcal{O}\left(\log\left(\frac{1}{\epsilon}\right)\right)$ iterations.

**Lemma A.11.** *If Lemma 5.4 is true, then $\exists \eta$, such that there is a constant factor decrease in $\|\boldsymbol{w}_2^{(t)}\|_2^2$ at each time step $t$, if we perform gradient ascent on the objective $\mathcal{M}_{\mathcal{D}}(\boldsymbol{w})$ starting from the initialization in Lemma 5.2. Thus, in $T = \mathcal{O}\left(\log\left(\frac{1}{\epsilon}\right)\right)$ iterations we get:*

$$|w_1^{(T)}| \geqslant \sqrt{1 - \epsilon^2}, \;\; \text{and} \;\; \|\boldsymbol{w}_2^{(T)}\|_2 \leqslant \epsilon.$$

*Proof:*

Since, $\tilde{\boldsymbol{w}}^{(t+1)} = \boldsymbol{w}^{(t)} + \eta \cdot \nabla_{\boldsymbol{w}^{(t)}} \mathcal{M}_{\mathcal{D}}(\boldsymbol{w}^{(t)})$:

$$
\begin{aligned}
\|\tilde{\boldsymbol{w}}_2^{(t+1)}\|_2^2 &= \|\boldsymbol{w}_2^{(t)} + \eta \nabla_{\boldsymbol{w}_2^{(t)}} \mathcal{M}_{\mathcal{D}}(\boldsymbol{w}^{(t)})\|_2^2 \\
&= \|\boldsymbol{w}_2^{(t)}\|_2^2 + \eta^2 \|\nabla_{\boldsymbol{w}_2^{(t)}} \mathcal{M}_{\mathcal{D}}(\boldsymbol{w}^{(t)})\|_2^2 + 2\eta \langle \nabla_{\boldsymbol{w}_2^{(t)}} \mathcal{M}_{\mathcal{D}}(\boldsymbol{w}^{(t)}), \boldsymbol{w}_2^{(t)} \rangle \\
&\leqslant (1 - 2\eta c_1) \cdot \|\boldsymbol{w}_2^{(t)}\|_2^2 + \eta^2 \|\nabla_{\boldsymbol{w}_2^{(t)}} \mathcal{M}_{\mathcal{D}}(\boldsymbol{w}^{(t)})\|_2^2, \qquad c_1 > 0 \text{ from Lemma 5.4}
\end{aligned}
$$

In Lemma A.10 we had identified that $\|\nabla_{\boldsymbol{w}_2^{(t)}} \mathcal{M}_{\mathcal{D}}(\boldsymbol{w}^{(t)})\|_2^2 \leqslant {c_1'}^2 \|\boldsymbol{w}_2^{(t)}\|_2^2$ for some $c_1' > 0$ using the expression in Equation 28. Thus,

$$\|\tilde{\boldsymbol{w}}_2^{(t+1)}\|_2^2 \;\leqslant\; (1 - 2\eta c_1 + \eta^2 {c_1'}^2) \cdot \|\boldsymbol{w}_2^{(t)}\|_2^2 \tag{31}$$

Thus, for a specific choice of $\eta$, one where $(1 - 2\eta c_1 + \eta^2 {c_1'}^2) < 1$, we get:

$$\|\tilde{\boldsymbol{w}}_2^{(t+1)}\|_2^2 \leqslant \kappa \cdot \|\boldsymbol{w}_2^{(t)}\|_2^2, \qquad \kappa < 1. \tag{32}$$

Similarly, from $w_1$ update: $\|\tilde{w}_1^{(t+1)}\|_2^2 = \|w_1^{(t)}\|_2^2 + \eta^2 \|\frac{\partial \mathcal{M}_{\mathcal{D}}(\boldsymbol{w}^{(t)})}{\partial w_1^{(t)}}\|_2^2 + 2\eta \langle \frac{\partial \mathcal{M}_{\mathcal{D}}(\boldsymbol{w}^{(t)})}{\partial w_1^{(t)}}, w_1^{(t)} \rangle$. Since $\langle \frac{\partial \mathcal{M}_{\mathcal{D}}(\boldsymbol{w}^{(t)})}{\partial w_1^{(t)}}, w_1^{(t)} \rangle > 0$, we get:

$$|\tilde{w}_1^{(t+1)}| \;>\; |w_1^{(t)}| \tag{33}$$

Recall that we re-normalize $\tilde{\boldsymbol{w}}^{(t+1)}$, i.e., $\boldsymbol{w}^{(t+1)} = \tilde{\boldsymbol{w}}^{(t+1)}/\|\tilde{\boldsymbol{w}}^{(t+1)}\|_2$. Hence, from Equation 32 and Equation 33 we can conclude: $\|\boldsymbol{w}_2^{(t+1)}\|_2^2 = (1 - \Omega(1)) \cdot \|\boldsymbol{w}_2^{(t)}\|_2^2$, a constant factor reduction. With a telescoping argument on $\|\boldsymbol{w}_2^{(t+1)}\|_2^2$, we would need $T = \mathcal{O}(\log(1/\epsilon))$ iterations for $\|\boldsymbol{w}_2^{(T)}\|_2^2 \leqslant \epsilon^2$. This completes the proof.

Lemma A.11 furnishes the guarantees in Theorem 5.3 when we perform project gradient ascent with respect to the population objective $\mathcal{M}_{\mathcal{D}}(\boldsymbol{w})$. We discussed this setting first since it captures the main intuition and simplifies some of the calculation. In the next subsection we shall look at the finite sample case, where RCAD optimizes $\mathcal{M}_{\hat{\mathcal{D}}}(\boldsymbol{w})$.

### A.4.1 Revisiting Lemma A.11 in a finite sample setting

Note that Lemma A.11 furnished guarantees when we took projected gradient ascent steps on the population objective $\mathcal{M}_{\mathcal{D}}(\boldsymbol{w})$. In fact, it gave us $\lim_{T \to \infty} \|\boldsymbol{w}_2^{(T)}\|_2 = 0$ and $\lim_{T \to \infty} w_1^{(T)} = 1$. But, in practice, we only have access to gradients of the finite sample objective $\mathcal{M}_{\hat{\mathcal{D}}}(\boldsymbol{w})$, where the expectation is taken with respect to the empirical measure over training data $\hat{\mathcal{D}}$.

We still follow the same arguments that we used to prove Lemma A.11, but now with sample gradients that closely approximate the true gradients through some uniform convergence arguments over the random variable $\nabla_{\boldsymbol{w}} \mathcal{M}_{\hat{\mathcal{D}}}(\boldsymbol{w})$. Then the monotonic results in Lemma 5.4 hold with high probability if we are given sufficient training data, i.e., $n$ is large enough that $\nabla_{\boldsymbol{w}} \mathcal{M}_{\hat{\mathcal{D}}}(\boldsymbol{w}) \approx \nabla_{\boldsymbol{w}} \mathcal{M}_{\mathcal{D}}(\boldsymbol{w})$, $\forall \boldsymbol{w}$, along directions $w_1$ and $\boldsymbol{w}_2$. To formalize this notion, we rely on an adapted form of Theorem C.1 in Chen et al. [7]. Although the conditions in our adaptation (Theorem A.12) are slightly different, the proof technique is identical to that of Theorem C.1 in [7]. Hence, we only present the key ideas needed to ensure that finite sample approximation does not change the behavior in of $\boldsymbol{w}^{(t)}$ we saw in Lemma 5.4, and refer interested readers to [7] for more details.

**Theorem A.12** (finite sample approximation from Chen et al. [7])**.** *Assume that* $\langle \frac{\partial \mathcal{M}_{\mathcal{D}}(\boldsymbol{w})}{\partial w_1}, w_1 \rangle > 0$, $\langle \nabla_{\boldsymbol{w}_2} \mathcal{M}_{\mathcal{D}}(\boldsymbol{w}), \boldsymbol{w}_2 \rangle < -c_1 \cdot \|\boldsymbol{w}_2\|_2^2$, *and* $\|\nabla_{\boldsymbol{w}_2} \mathcal{M}_{\mathcal{D}}(\boldsymbol{w})\|_2^2 \leqslant {c_1'}^2 \|\boldsymbol{w}_2\|_2^2$, *for constants* $c_1, c_1' > 0$. *If* $n = \tilde{\mathcal{O}}(\frac{d \|\tilde{\Sigma}\|_{\mathrm{op}}}{\epsilon^2} \log(1/\delta))$ *where* $\epsilon = \mathcal{O}(c_1 \tau^2)$ *for some small* $\tau < 0.5$, *then with probability* $1 - \delta$ *over* $\hat{\mathcal{D}}$, *the empirical and true gradients along* $w_1$ *and* $\boldsymbol{w}_2$ *are close:*

$$|\langle \nabla_{w_1} \mathcal{M}_{\mathcal{D}}(\boldsymbol{w}) - \nabla_{w_1} \mathcal{M}_{\hat{\mathcal{D}}}(\boldsymbol{w}), w_1 \rangle| \leqslant \epsilon$$
$$|\langle \nabla_{\boldsymbol{w}_2} \mathcal{M}_{\mathcal{D}}(\boldsymbol{w}) - \nabla_{\boldsymbol{w}_2} \mathcal{M}_{\hat{\mathcal{D}}}(\boldsymbol{w}), \boldsymbol{w}_2 \rangle| \leqslant \epsilon \tag{34}$$

*Furthermore, if the true feature at initialization is dominant i.e.,* $|w_1^{(0)}| \geqslant 0.5$ *and if* $\|\boldsymbol{w}_2^{(t)}\|_2 \geqslant \tau/2$, *then for* $\eta = \mathcal{O}(c_1/{c_1'}^2)$ *the norm along* $\boldsymbol{w}_2$ *shrinks by a constant factor:* $\|\boldsymbol{w}_2^{(t+1)}\|_2^2 \leqslant (1 - \mathcal{O}(c_1^2/{c_1'}^2))\|\boldsymbol{w}_2^{(t)}\|_2^2$. *This continues until* $\|\boldsymbol{w}_2^{(t)}\|_2 < \tau/2$ *after which it stabilizes and* $\|\boldsymbol{w}_2^{(t)}\|_2 \leqslant \tau$ *is guaranteed in subsequent iterations.*

In Theorem A.12 we allow for some error $\epsilon$ in approximating the true gradient with the finite sample approximation along directions $w_1$ and $\boldsymbol{w}_2$. Note, that this error is much smaller $O(c_1 \tau^2)$, than the final error tolerated on the norm of $\boldsymbol{w}_2$ which is $\tau$, when $\tau < 0.5$. Essentially, using Lemma C.1 in [7], it is easy to show that Equation 34 holds with probability $1 - \delta$ if $n = \tilde{\mathcal{O}}(\frac{d \|\tilde{\Sigma}\|_{\mathrm{op}}}{\epsilon^2} \log(1/\delta))$, where $\tilde{\mathcal{O}}$ hides logarithmic factors in $1/\epsilon^2$. This, allows us to make the following replacements for our expressions in Lemma A.11 which considered the population version:

$$\langle \frac{\partial \mathcal{M}_{\mathcal{D}}(\boldsymbol{w})}{\partial w_1}, w_1 \rangle > 0 \quad \longleftrightarrow \quad \langle \frac{\partial \mathcal{M}_{\hat{\mathcal{D}}}(\boldsymbol{w})}{\partial w_1}, w_1 \rangle > -\epsilon$$
$$\langle \nabla_{\boldsymbol{w}_2} \mathcal{M}_{\mathcal{D}}(\boldsymbol{w}), \boldsymbol{w}_2 \rangle < -c_1 \cdot \|\boldsymbol{w}_2\|_2^2 \quad \longleftrightarrow \quad \langle \nabla_{\boldsymbol{w}_2} \mathcal{M}_{\hat{\mathcal{D}}}(\boldsymbol{w}), \boldsymbol{w}_2 \rangle < -c_1 \cdot \|\boldsymbol{w}_2\|_2^2 + \epsilon$$
$$\|\nabla_{\boldsymbol{w}_2} \mathcal{M}_{\mathcal{D}}(\boldsymbol{w})\|_2^2 \leqslant {c_1'}^2 \|\boldsymbol{w}_2\|_2^2 \quad \longleftrightarrow \quad \|\nabla_{\boldsymbol{w}_2} \mathcal{M}_{\hat{\mathcal{D}}}(\boldsymbol{w})\|_2^2 \leqslant 2{c_1'}^2 \|\boldsymbol{w}_2\|_2^2 + 2\epsilon^2$$

Thus, when $\eta = \mathcal{O}\left(\frac{c_1}{{c_1'}^2}\right)$, there is a constant factor decrease in $\|\tilde{\boldsymbol{w}}\|_2^2$:

$$\|\tilde{\boldsymbol{w}}_2^{(t+1)}\|_2^2 \leqslant (1 - \mathcal{O}(c_1^2/{c_1'}^2))\|\boldsymbol{w}_2^{(t)}\|_2^2 + 2\eta^2 \epsilon^2 + 2\eta\epsilon$$

On the other hand since $\eta\epsilon$ is small, there is a significant increase in $|\tilde{w}_1|$:

$$\|\tilde{w}_1^{(t+1)}\|_2^2 \geqslant \|w^{(t)}\|_2^2 - 2\eta\epsilon$$

Thus, any decrease in $|w_1|$ is relatively much smaller compared to decrease in $\|\boldsymbol{w}_2\|_2$. After renormalization we get:

$$\|\boldsymbol{w}_2^{(t+1)}\|_2^2 \leqslant (1 - \mathcal{O}(c_1^2/{c_1'}^2))\|\boldsymbol{w}_2^{(t)}\|_2^2$$

Given the constant factor reduction, the number of iterations needed to drive $\|\boldsymbol{w}_2^{(t+1)}\|_2$ to be less than $\tau$ is $\mathcal{O}(\log(1/\tau))$. Furthermore, in the finite sample case, we need $n = \tilde{\mathcal{O}}(\frac{1}{\tau^4} \log(1/\delta))$ samples for the $\epsilon$−approximation in Equation 34 to hold where $\epsilon = \mathcal{O}(c_1 \tau^2)$. We can now combine this requirement on the sample size with the lower bound on $n$ from Corollary A.7 needed to maintain the initialization conditions in Lemma 5.2 with $\beta w_1 \geqslant K\alpha$, to get $n \gtrsim \frac{\beta^2 + \log(1/\delta) \cdot \|\tilde{\Sigma}\|_{\mathrm{op}} + \|\tilde{\Sigma}\|_*}{\gamma^2 \mathrm{erfc}\left(K\alpha/\sqrt{2\sigma_{\min}(\tilde{\Sigma})}\right)^2} + \frac{\log 1/\delta}{\tau^4}$.

This completes the proof for Theorem 5.3.

# B   Additional Details for Experiments in Section 4

In this section, we first present technical details useful for reproducing our empirical results in Section 4. We also take a note of the computational resources needed to run the experiments in this paper. Then we present some tables containing absolute test performance values for some of the plots in Section 4 where we plotted the relative performance of each method with respect to empirical risk minimization (ERM). Finally, we present some preliminary results on Imagenet classification benchmark.

**Hyperparameter details.** The hyperparameters used to train ADA [62], ME-ADA [73], and adversarial training (FGSM) [14] are identical to those reported in the original works we cite. For label

smoothing, we tune the smoothing parameter on the validation set and find $\epsilon = 0.6$ to be optimal for the smaller dataset of CIFAR-100-2k, and $\epsilon = 0.2$ yields the best results on other datasets. For RCAD we have two hyperparameters $\alpha, \lambda$ and find that $\alpha = 0.5, \lambda = 0.02$ gives good performance over all benchmarks except CIFAR-100-2k for which we use $\alpha = 1.0, \lambda = 0.1$.

Unless specified otherwise, we train all methods using the ResNet-18 [19] backbone, and to accelerate training loss convergence we clip gradients in the $l_2$ norm (at 1.0) [71, 18]. We train all models for 200 epochs and use SGD with an initial learning rate of 0.1 and Nesterov momentum of 0.9, and decay the learning rate by a factor of 0.1 at epochs 100, 150 and 180 [10]. We select the model checkpoint corresponding to the epoch with the best accuracy on validation samples as the final model representing a given training method. For all datasets (except CIFAR-100-2k and CIFAR-100-10k for which we used 32 and 64 respectively) the methods were trained with a batch size of 128. For our experiments involving the larger network of Wide ResNet 28-10, we use the optimization hyperparameters mentioned in Zagoruyko and Komodakis [67].

**Computational resources.** To run any experiment we used at most two NVIDIA GEFORCE GTX 1080Ti GPU cards. As we mention in the main paper, any run of our method RCAD takes at most 30% more per-batch processing time than ERM with data augmentation on a given dataset. All runs of RCAD with ResNet-18 backbone on CIFAR-100-2k, CIFAR-100-10k, CIFAR-100, CIFAR-10 and SVHN take less than 20 hours, while Tiny Imagenet takes about 36 hours. As we mention in Section 6, one of the main limitations of RCAD is that it increases the running time by 30% compared to ERM training. But RCAD training is much faster than alternative adversarial data augmentation baselines ADA and ME-ADA which increase running time by 160% and 185% (over ERM), respectively.

**Absolute test performance values.** In Table 2, we plot the test performance values for our method RCAD which maximizes entropy on self-generated perturbations, and adversarial baselines which minimize cross-entropy on adversarial examples. The plot in Figure 3a is derived from the results in Table 2. Table 3a compares the performance of RCAD with the most competitive adversarial baseline ME-ADA, when both methods are used in combination with the label smoothing regularizer. These results are also presented in Figure 3a and discussed in Section 4.2. For the plot in Figure 4a comparing baselines ERM and ME-ADA to our method RCAD with the larger network Wide ResNet 28-10 [67], we show the test accuracy values on CIFAR-100(-2k/10k) in Table 3b. Finally, Table 4 presents the test accuracy of each method on the CIFAR-100 test set, when trained on training datasets of varying sizes. Figure 3b plots the same results relative to ERM. The trends observed here and in Figure 3b are discussed in Section 4.3. In all the above, we show the mean accuracy and 95% confidence interval evaluated over 10 independent runs.

| Dataset | ERM | FGSM | ADA | ME-ADA | RCAD |
|---|---|---|---|---|---|
| CIFAR-100-2k | $28.5 \pm 0.12$ | $27.1 \pm 0.08$ | $28.7 \pm 0.12$ | $29.4 \pm 0.11$ | $\mathbf{30.4 \pm 0.09}$ |
| CIFAR-100-10k | $60.4 \pm 0.09$ | $57.6 \pm 0.09$ | $60.0 \pm 0.10$ | $60.4 \pm 0.08$ | $\mathbf{61.9 \pm 0.06}$ |
| Tiny Imagenet | $66.4 \pm 0.07$ | $62.4 \pm 0.07$ | $66.5 \pm 0.08$ | $66.9 \pm 0.10$ | $\mathbf{67.3 \pm 0.07}$ |
| CIFAR-10 | $95.2 \pm 0.04$ | $93.1 \pm 0.04$ | $95.2 \pm 0.03$ | $\mathbf{95.6 \pm 0.05}$ | $\mathbf{95.7 \pm 0.04}$ |
| CIFAR-100 | $76.4 \pm 0.05$ | $73.2 \pm 0.05$ | $76.5 \pm 0.05$ | $77.1 \pm 0.07$ | $\mathbf{77.3 \pm 0.05}$ |
| SVHN | $97.5 \pm 0.02$ | $95.4 \pm 0.02$ | $97.5 \pm 0.02$ | $97.4 \pm 0.04$ | $\mathbf{97.6 \pm 0.02}$ |

Table 2: **RCAD consistently outperforms adversarial baselines.** In Section 4.2 we investigated the performance of RCAD against baseline methods that use the adversarially generated perturbations differently, i.e., methods that reduce cross-entropy loss on adversarial examples. This table presents the test accuracies for RCAD and other baselines with 95% confidence interval computed over 10 independent runs. Figure 3a plots the results reported here by looking at the performance improvement of each method over the ERM baseline.

**Preliminary results on Imagenet classification.** We take a ResNet-50 [19] model pretrained on Imagenet [9] and measure its *top-1* and *top-5* test accuracy. Next, we tune this pretrained ResNet-50 model, using the optimization hyperparameters from Yun et al. [66]. We refer to the model produced by this method as "Finetuned with ERM". We compare the test accuracies for these two models on the Imagenet test set with the test accuracy of the model that is obtained by finetuning the pretrained ResNet-50 model with the RCAD objective in Equation 2 ("Finetuned with RCAD"). Results for this experiment can be found in Table 5. We do not finetune RCAD hyperparameters $\alpha, \lambda$ on Imagenet, and retain the ones used for the TinyImagenet benchmark. We faced issues running experiments for the competitive but expensive adversarial baseline ME-ADA since the GPU memory needed to run ME-ADA on Imagenet exceeded the limit on our GPU cards. Mainly for these two reasons,

| Dataset | ME-ADA+LS | RCAD+LS |
|---|---|---|
| CIFAR-100-2k | $30.5 \pm 0.13$ | $\mathbf{32.2 \pm 0.11}$ |
| CIFAR-100-10k | $61.15 \pm 0.07$ | $\mathbf{63.1 \pm 0.08}$ |
| Tiny Imagenet | $67.1 \pm 0.08$ | $\mathbf{67.9 \pm 0.07}$ |
| CIFAR-10 | $\mathbf{95.65 \pm 0.04}$ | $\mathbf{95.7 \pm 0.03}$ |
| CIFAR-100 | $77.8 \pm 0.06$ | $\mathbf{79.1 \pm 0.06}$ |
| SVHN | $97.42 \pm 0.04$ | $\mathbf{97.6 \pm 0.02}$ |

(a)

| Dataset | ERM | ME-ADA | RCAD |
|---|---|---|---|
| C100-2k | $30.2 \pm 0.10$ | $31.8 \pm 0.07$ | $\mathbf{32.3 \pm 0.08}$ |
| C100-10k | $62.5 \pm 0.07$ | $62.6 \pm 0.08$ | $\mathbf{62.9 \pm 0.05}$ |
| C100 | $81.2 \pm 0.07$ | $81.0 \pm 0.07$ | $\mathbf{81.4 \pm 0.06}$ |

(b)

Table 3: *(Left)* **RCAD outperforms adversarial baselines even when combined with existing regularizers.** We compare the performance of RCAD and ME-ADA when both methods are trained with label smoothing (smoothing parameter $\epsilon = 0.2$). We find that label smoothing improves the performance of both methods, but the benefit of RCAD still persists. Figure 3a also plots the same results, relative to the performance of ERM. *(Right)* **RCAD can outperform baselines when used with larger backbones.** Here, we train our method RCAD and baselines ERM, ME-ADA with the larger architecture Wide ResNet 28-10 on CIFAR-100 (C100), CIFAR-100-2k (C100-2k) and CIFAR-100-10k (C100-10k) benchmarks. We find that while the performance gains of RCAD over ME-ADA and ERM are higher with the ResNet-18 backbone, RCAD continues to outperform them here as well. Figure 4a also plots the same results, relative to the performance of ERM.

| Dataset size | ERM | LS | FGSM | ME-ADA | RCAD |
|---|---|---|---|---|---|
| 500 | $11.9 \pm 0.06$ | $13.5 \pm 0.08$ | $13.2 \pm 0.06$ | $13.1 \pm 0.06$ | $\mathbf{14.8 \pm 0.08}$ |
| 1,000 | $18.9 \pm 0.07$ | $19.7 \pm 0.07$ | $18.4 \pm 0.08$ | $18.6 \pm 0.08$ | $\mathbf{21.3 \pm 0.05}$ |
| 2,000 | $28.5 \pm 0.05$ | $29.2 \pm 0.07$ | $27.1 \pm 0.07$ | $29.4 \pm 0.05$ | $\mathbf{30.4 \pm 0.05}$ |
| 5,000 | $52.2 \pm 0.07$ | $52.7 \pm 0.06$ | $50.9 \pm 0.07$ | $51.4 \pm 0.06$ | $\mathbf{53.6 \pm 0.05}$ |
| 10,000 | $60.4 \pm 0.07$ | $61.5 \pm 0.06$ | $57.6 \pm 0.06$ | $60.4 \pm 0.07$ | $\mathbf{61.9 \pm 0.06}$ |
| 50,000 | $76.4 \pm 0.07$ | $77.4 \pm 0.08$ | $73.2 \pm 0.08$ | $77.1 \pm 0.07$ | $\mathbf{77.7 \pm 0.06}$ |

Table 4: **RCAD is more effective than other existing regularizers in low data regime.** We evaluate the effectiveness of RCAD over other baseline regularizers as we decrease the size of the training data for CIFAR-100. We find that the performance gain from using RCAD is higher as training data size reduces, possibly since RCAD is more robust to spurious correlations in the training data (compared to other baselines), and deep models are more vulnerable to such correlations in the low data regime [68]. Figure 3b also plots the same results, relative to the performance of ERM.

we note that our results on the Imagenet benchmark are preliminary. Nevertheless, the results are promising since we see an improvement in test accuracy when we finetune the pretrained model with our RCAD objective as opposed to the standard cross-entropy loss as part of the ERM objective. The improvement is more significant for top-1 test accuracy.

| Test Accuracy | Pretrained ResNet-50 | Finetuned with ERM | Finetuned with RCAD |
|---|---|---|---|
| top-1 | $69.81 \pm 0.07\%$ | $77.45 \pm 0.05\%$ | $77.76 \pm 0.06\%$ |
| top-5 | $89.24 \pm 0.06\%$ | $93.21 \pm 0.04\%$ | $93.29 \pm 0.05\%$ |

Table 5: **Preliminary results on Imagenet classification.** We compare the performance of a pretrained ResNet-50 model, with the models obtained by finetuning the pretrained model with either the cross-entropy loss (finetuned with ERM) or the RCAD objective (finetuned with RCAD). We show the mean accuracy and $95\%$ confidence interval evaluated over 7 independent runs.

**Comparisons with Virtual Adversarial Training [37].** In Section 4.3 we find that RCAD is most effective in low-data regimes. Hence, we would like to compare it with semi-supervised methods that regularize models trained on a small fraction of labeled data, with unsupervised objectives on a larger set of unlabeled data. In the context of using adversarially generated examples, we compare RCAD with Virtual Adversarial Training [37] (VAT) which makes the model's predictive distribution robust to small perturbations on a given input. Note that the VAT regularizer does not require supervision (true label). For a fair comparison, we adapt RCAD to the semi-supervised setting with the objective in Equation 35, where the labeled and unlabeled sets of data points are denoted as $\hat{\mathcal{D}}_l$ and $\hat{\mathcal{D}}_u$ respectively. In addition to optimizing our objective in Equation 2 on $\hat{\mathcal{D}}_l$, we optimize the entropy term on the unlabeled points in $\hat{\mathcal{D}}_u$. For each example in $\hat{\mathcal{D}}_u$, we simply maximize entropy on its corresponding

|  | CIFAR-100-2k | CIFAR-100-10k |
|---|---|---|
| VAT [37] | $32.74 \pm 0.06\%$ | $62.81 \pm 0.06\%$ |
| RCAD | $33.03 \pm 0.05\%$ | $62.98 \pm 0.05\%$ |

Table 6: **Comparing RCAD with semi-supervised baseline in low-data regime:** On CIFAR-100-2k and CIFAR-100-10k we train with the semi-supervised algorithm Virtual Adversarial Training (VAT) [37] and compare the test performance with a model that is trained with the semi-supervised form of the RCAD objective (Equation 35). The 95% confidence intervals are obtained by evaluating the test performance of models from 10 independent runs for both methods.

OOD example generated using Equation 1 with the main difference being that we treat the model's predicted label ($\hat{y}$) as the true label, similar to what VAT does when generating adversarial examples on unlabeled data. We run VAT and semi-supervised RCAD on our subsampled CIFAR-100 training sets: CIFAR-100-2k and CIFAR-100-10k with 2000 and 10,000 labeled training samples respectively. The rest of the CIFAR-100 training data is treated as the unlabeled set. Our findings in Table 6 indicate that RCAD improves over VAT by 0.29% and 0.17% on CIFAR-100-2k and CIFAR-100-10k respectively.

$$
\begin{aligned}
\arg\max_{\boldsymbol{w} \in \mathcal{W}} \quad & \mathbb{E}_{\hat{\mathcal{D}}_l} \left[ \log p_{\boldsymbol{w}}(y \mid \mathbf{x}) + \lambda \cdot \mathcal{H}_{\boldsymbol{w}} \left( \mathbf{x} - \alpha \cdot \nabla_{\mathbf{x}} \log p_{\boldsymbol{w}}(\mathbf{y} \mid \mathbf{x}) \right) \right] \\
& + \lambda \cdot \mathbb{E}_{\hat{\mathcal{D}}_u} \left[ \mathcal{H}_{\boldsymbol{w}} \left( \mathbf{x} - \alpha \cdot \nabla_{\mathbf{x}} \log p_{\boldsymbol{w}}(\hat{\mathbf{y}} \mid \mathbf{x}) \right) \right] \\
\text{where,} \quad \hat{\mathbf{y}} \;=\; & \arg\max_{y \in \mathcal{Y}} \; p_{\boldsymbol{w}}(y \mid \mathbf{x})
\end{aligned}
\tag{35}
$$

In order to train VAT, we used the original paper authors' implementation and set hyperparameters $\alpha, \epsilon$ (see Equations 5, 8 in Miyato et al.) to $\alpha = 1.0, \epsilon = 8.0$ for both datasets. RCAD hyperparameters were set to $\alpha = 0.7, \lambda = 0.1$ for CIFAR-100-2k and $\alpha = 0.5, \lambda = 0.02$ for CIFAR-100-10k. For both methods, the respective hyperparameters were tuned using a hold out validation set.

## C    RCAD can Improve Robustness to Adversarial/Natural Distribution Shifts

Typically, methods for handling distribution shift are different from methods for improving test accuracy. Prior work has found that increasing robustness to distribution shifts tends to be at odds with increasing test accuracy: methods that are more robust often achieve lower test accuracy, and methods that achieve higher test accuracy tend to be less robust [48, 70, 61]. While the main aim of our experiments is to show that RCAD improves test accuracy, our next set of experiments investigate whether RCAD is any more robust to distribution shift than baseline methods.

For these experiments on distribution shift, we use the exact same hyperparameters as in the previous experiments. Better results are likely achievable by tuning the method for performance on these robustness benchmarks. By reporting results using the exact same hyperparameters, we demonstrate that the same method might both achieve high in-distribution performance and out-of-distribution performance.

| Method | CIFAR-100-2k | | CIFAR-100 | | CIFAR-10 | |
|---|---|---|---|---|---|---|
| | Clean | FGSM Attack | Clean | FGSM Attack | Clean | FGSM Attack |
| ERM | $28.5 \pm 0.12$ | $24.7 \pm 0.10$ | $76.4 \pm 0.05$ | $67.4 \pm 0.07$ | $95.2 \pm 0.04$ | $88.3 \pm 0.03$ |
| Adv. Training | $27.1 \pm 0.08$ | $27.2 \pm 0.08$ | $73.2 \pm 0.05$ | $73.1 \pm 0.04$ | $93.1 \pm 0.04$ | $92.9 \pm 0.04$ |
| ADA | $28.7 \pm 0.12$ | $27.0 \pm 0.08$ | $76.5 \pm 0.05$ | $72.7 \pm 0.05$ | $95.2 \pm 0.03$ | $88.1 \pm 0.04$ |
| ME-ADA | $29.4 \pm 0.11$ | $27.6 \pm 0.11$ | $77.1 \pm 0.07$ | $\mathbf{74.8 \pm 0.03}$ | $\mathbf{95.6 \pm 0.05}$ | $\mathbf{93.1 \pm 0.04}$ |
| RCAD | $\mathbf{30.4 \pm 0.09}$ | $\mathbf{28.1 \pm 0.10}$ | $\mathbf{77.3 \pm 0.05}$ | $74.5 \pm 0.04$ | $\mathbf{95.7 \pm 0.04}$ | $93.0 \pm 0.03$ |

Table 7: **Robustness to adversarial perturbations compared against in-distribution test accuracy:** Clean (evaluation on *iid* test data) and Robust test accuracies of adversarial methods (adversarial training (FGSM [14]), ADA and ME-ADA), ERM and RCAD when the adversary carries out attacks using the Fast Gradient Sign (FGSM) method when the strength of the attack in $l_1$ norm is 0.05. We show the mean accuracy and 95% confidence interval evaluated over 10 independent runs.

## C.1 Robustness to adversarial attacks

We first look at robustness to adversarial attacks, using FGSM [14] as the attack method. The conventional approach to fending off adversarial attacks is adversarial training, wherein the training objective exactly matches the testing objective. Thus, adversarial training represents the "gold standard" for performance on this evaluation. We compare the adversarial robustness of RCAD, adversarial training, ADA, and ME-ADA in Table 7. Not only does our method achieve higher (clean) test accuracy than adversarial training on all datasets, but surprisingly it also achieves higher robust test accuracy on the harder CIFAR-100-2k benchmark where the clean test accuracy of RCAD is $+3\%$ greater than adversarial training, and robust test accuracy is $+0.5\%$ better than ME-ADA. Although, we find that adversarial baselines achieve better robust test accuracy when the training data is sufficient (e.g., CIFAR-100 and CIFAR-10 in Table 7).

Both ADA and ME-ADA perform some form of adversarial training, so it is not surprising that they outperform RCAD on this task. We suspect that these methods outperform adversarial training because they are trained using the multi-step projected gradient descent (PGD) [36], rather than the one-step FGSM [14] and also have a higher clean test accuracy. While our aim is *not* to propose a state-of-the-art method for withstanding adversarial attacks, these preliminary results suggest that RCAD may be somewhat robust to adversarial attacks, but does so without inheriting the poor (clean) test accuracy of standard adversarial training.

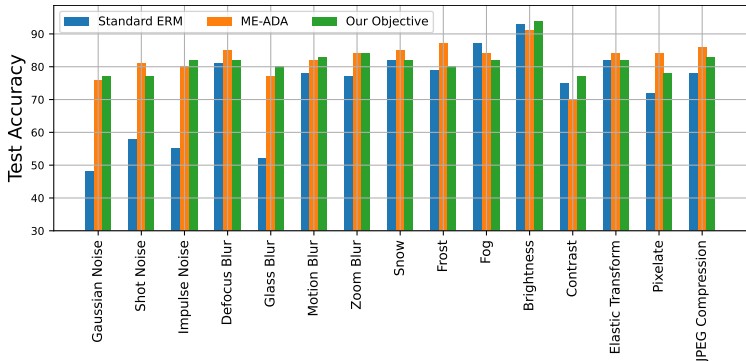

Figure 6: **Robustness to natural shifts in distribution:** Plots comparing the performance of standard ERM training and ME-ADA against our method (RCAD) trained on CIFAR-10 benchmark and tested on various distribution shifts in the corrupted CIFAR-10-C benchmark.

## C.2 Robustness to natural distribution shifts

Our final set of experiments probe robustness to more systematic distribution shifts using the corrupted CIFAR-10-C dataset [21]. These shifts go beyond the small perturbations introduced by adversarial examples, and are a more faithful reflection of the sorts of perturbations a machine learning model might face "in the wild".

We compare RCAD to standard ERM and ME-ADA on this benchmark; when all methods are trained on the uncorrupted CIFAR-10 training dataset, but evaluated on different types of corruptions. We report results in Figure 6. Both RCAD and ME-ADA consistently outperform ERM. On certain corruptions (e.g., Gaussian noise, glass blur), RCAD and ME-ADA achieve test accuracies that are around $+25\%$ larger than the ERM baseline. We do not notice any systematic difference in the results of RCAD versus ME-ADA, but note that ME-ADA requires $2\times$ more compute than RCAD because its adversarial examples require multiple gradient steps to compute.

While the main aim of our experiments has been to show that RCAD achieves higher test accuracy, its good performance on robustness benchmarks suggests that it may be a simpler yet appealing choice for practitioners.

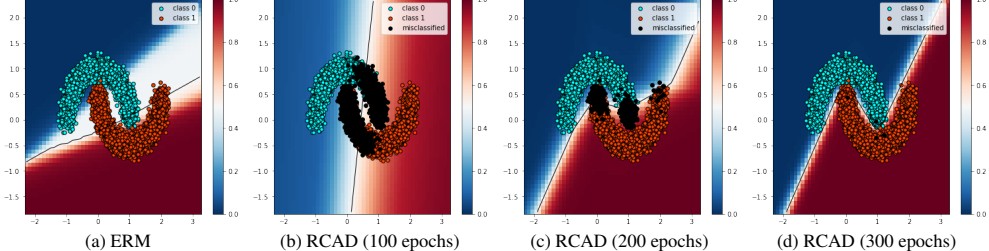

| (a) ERM | (b) RCAD (100 epochs) | (c) RCAD (200 epochs) | (d) RCAD (300 epochs) |

Figure 7: **RCAD objective learns decision boundary using only generalizable features:** We simulate a high-dimensional non-linear classification problem by projecting a simple 2-d dataset into a 625-dimensional space. *(a)* Standard ERM training overfits to this dataset, achieving perfect training accuracy by picking up on spurious features. Plotting a 2D projection of the decision boundary along the true features, we see that it poorly separates the data along the generalizable (true) features. *(b, c, d)* Visualizing our method (RCAD) at different snapshots throughout training, we see that it converges to the true decision boundary using only the generalizable features, thereby achieving a higher test accuracy.

## D  Analyzing RCAD vs. ERM in a Non-linear Toy Classification Setup

In Section 5 we analyse the solution returned by RCAD and compare it to the ERM solution in a simplified binary classification setting where the hypothesis class consists of linear predictors. We find that the high dimensional linear decision boundary learnt by ERM depends on the noisy dimensions that are spuriously correlated with the label on the training data $\hat{\mathcal{D}}$. On the other hand, our method RCAD corrects this decision boundary by unlearning spurious features. The self-generated examples by RCAD exemplify the spurious correlations, and maximizing entropy on these examples helps the model to drive any learnt components along the noisy dimensions to lower values. In this section, we consider a setup where the true decision boundary is non-linear and the hypothesis class consists of two-layer neural networks with 128 hidden units and ReLU activations. In order to induce spurious features in our training data, we stick to high-dimensional inputs.

Typically, in classification problems the true generalizable features span a lower dimensional space (compared to the ambient dimension) [1]. Ideally, we would like to learn classifiers that depend only on these few generalizable features, and remain independent of all the other noisy features. However, training neural networks with the cross-entropy loss and stochastic gradient descent (SGD) often leads to overfitting on training data since the trained model ends up fitting on noisy features with high probability [44, 20]. To simulate this phenomenon, we use a $d-$dimensional toy classification problem where the true features are given by the first two-dimensions only, while the rest of the $d - 2$ dimensions are pure noise.

A dataset $\hat{\mathcal{D}}$ of 20,000 training examples is generated by first sampling a two-dimensional vector $\tilde{\mathbf{x}}$ with equal probability from one of two classes supported over two different well separated moon shaped regions (see Figure 7). Here, $\tilde{\mathbf{x}}$ is a two dimensional vector with label $y \in \{0, 1\}$. In our setting, the input dimension $d = 625$. Now, in order to construct the final $d-$dimensional input $\mathbf{x}$ for the classifier using this two dimensional sample $\tilde{\mathbf{x}}$, we first append to each sample a vector of $(d-2 = 623)$ zeros, and then add add Gaussian noise $\epsilon \sim \mathcal{N}(0, \sigma^2 \boldsymbol{I}_d)$ where $\sigma = 0.1$. Note that the noise is added to all dimensions, including the dimensions of the true feature but our results would not change if we choose to only add noise to the last $d - 2$ dimensions, which would only make the problem easier.

$$\mathbf{x} \triangleq (\tilde{\mathbf{x}}, \underbrace{0, \cdots, 0}_{\text{d-2=623 zeros}}) + \epsilon, \qquad \epsilon \sim \mathcal{N}(\mathbf{0}, \sigma^2 \mathbf{I_d})$$

We will measure the in-distribution performance of a classifier, trained on $\hat{\mathcal{D}}$ and evaluated on fresh examples from the distribution defined above. In our toy setup, the first two dimensions of the data perfectly explain the class label; we call these dimensions the generalizable (true) features. We are interested in learning non-linear classifiers (two-layer neural nets) that successfully identify these generalizable features, while ignoring the remaining dimensions.

Models trained with the standard cross-entropy loss on a fixed dataset $\hat{\mathcal{D}}$ are liable to overfit [68]. A model can best reduce the (empirical) cross-entropy loss by learning features that span all dimensions,

| $(d, \sigma)$ | ERM | Adv. Training (FGSM) | ME-ADA | RCAD |
|---|---|---|---|---|
| $(625, 0.10)$ | 80.4% | 78.8% | 90.9% | **94.9**% |
| $(625, 0.75)$ | 77.8% | 77.9% | 84.4% | **89.6**% |
| $(1000, 0.10)$ | 74.3% | 72.0% | 83.5% | **87.1**% |

Table 8: **In the toy non-linear classification problem, RCAD is found to be more robust to spurious correlations in the training data.** We show the test performance of RCAD vs. baseline methods ERM, adversarial training (FGSM) and ME-ADA as we vary two parameters: $d$ which is the dimension of the input, and $\sigma$ which is the standard deviation of the noise added to every dimension.

including the spurious feature dimensions. Precisely, the model can continue to reduce the cross-entropy loss on some example $\mathbf{x}^{(i)}$ by further aligning some of the weights of its hidden units along the direction of the corresponding noise $\epsilon^{(i)}$, that is part of input $\mathbf{x}^{(i)}$.

We start by training a two-layer net on $\hat{\mathcal{D}}$. We train it for 300 epochs of SGD using the standard empirical cross entropy loss. This model achieves perfect training accuracy ($100\%$), but performs poorly on the validation set ($80.4\%$). To visualize the learned model, we project the decision boundary on the first two coordinates, the only ones that are truly correlated with the label on population data. The decision boundary for this model, shown in Figure 7a, is quite different from the true decision boundary. Rather than identifying the true generalizable features, this model has overfit to the noisy dimensions, which are perpendicular to the span of the true features. Training this model for more epochs leads to additional overfitting, further decreasing the test accuracy.

Next we train with our objective RCAD using the same training dataset. In addition to minimizing the standard empirical cross entropy loss, our method also maximizes the predictive entropy of the model on self-generated perturbations that lie along the adversarial directions. Intuitively, we expect that these new examples will be along the directions of the spurious features, exacerbating them. This is because adversarial directions have been shown to comprise of spurious features [23]. Thus, in training the model to be less confident on these examples, we signal the model to unlearn these spurious features. Applying RCAD to this dataset, we achieve a much larger test accuracy of $94.9\%$. When we visualize the decision boundary (along span of the true features) in Figure 7d, we observe that it correctly separates the data. While SGD is implicitly biased towards learning simple (e.g., linear) decision boundaries [28], our results show that RCAD partially counters this bias, forcing the model to learn a non-linear decision boundary along the true features and ignoring the noisy dimensions. Additionally we also train adversarial baselines in this setup and compare the test accuracies with RCAD in Table 8. We find that RCAD is less vulnerable to spurious correlations compared to other methods even when we increase the dimensionality of the input or the noise level in the data generating process, both of which would make models more prone to overfitting. This is confirmed by the significant drop in the test performance of the ERM baseline.