# OpenReview forum: "Adversarial Unlearning: Reducing Confidence Along Adversarial Directions"
_NeurIPS.cc/2022/Conference — NeurIPS 2022 Accept_

### Official Review · Reviewer_6c8K · 2022-06-27

**Rating:** 7
**Confidence:** 3
**Soundness:** 4 excellent
**Presentation:** 4 excellent
**Contribution:** 4 excellent

**Summary:**

This paper proposes to increase the test accuracy by maximizing the entropy of out-of-distribution examples that are adversarially generated, which is named reducing confidence along adversarial directions (RCAD). Empirical results support the claim that RCAD can improve test accuracy. In addition,  the authors theoretically analyze the effectiveness of RCAD on a linear case.


**Questions:**

+ Could I know the influence of the hyperparameter $\lambda$, since it seems that an important hyperparameter in RCAD?


**Limitations:**

The authors have pointed out the limitation of their proposed method. I have no idea about how to accelerate RCAD so far.


**Strengths And Weaknesses:**

Strengths
+ This paper is well written and organized. I can easily follow.
+ The proposed method is easy but effective in improving test accuracy. The proposed method is an interesting case that leverages adversarial attacks for good.
+ Experiments are comprehensive and sound. The results validate that RCAD can be compatible with various methods and indeed enhance the test accuracy.
+ The theoretical results explain why RCAD help test accuracy to some extent (although I have not carefully checked the correctness of theoretical results) — RCAD can regularize the model to unlearn the spurious features.

---

> ### Author Response · Authors · 2022-08-02
> **Response to Reviewer 6c8K**
>
> We thank the reviewer for their detailed feedback and comments.
>
> > Influence of hyper-parameter $\lambda$
>
> RCAD is generally quite robust to the choice of the hyperparameter $\lambda$ (compared to $\alpha$), with values in [0.07, 0.12] all producing test performance significantly better than ERM. The table below shows that for CIFAR-100-2k the performance varies from 28.5% to 30.6%, which is a shorter range compared to the performance range (27.0% to 30.5%) observed when ablating on $\alpha$ in Fig 3b. When the dataset size is large we find that a much smaller value of $\lambda=0.02$ is sufficient since not much regularization is needed (Appendix B, L961).
>
> | $\lambda$ | Test Accuracy on CIFAR-100-2k |
> |----------:|:-----------------------------:|
> |      0.00 |       $28.5 \pm 0.12$%      |
> |      0.01 |       $28.5 \pm 0.09$%      |
> |      0.02 |       $28.7 \pm 0.10$%      |
> |      0.03 |       $28.6 \pm 0.11$%      |
> |      0.04 |       $29.2 \pm 0.12$%      |
> |      0.05 |       $29.3 \pm 0.11$%      |
> |      0.06 |       $29.8 \pm 0.08$%      |
> |      0.07 |       $29.6 \pm 0.09$%      |
> |      0.08 |       $30.2 \pm 0.10$%      |
> |      0.09 |       $30.5 \pm 0.08$%      |
> |      0.10 |       $30.4 \pm 0.09$%      |
> |      0.11 |       $30.6 \pm 0.07$%      |
> |      0.12 |       $30.3 \pm 0.08$%      |
> |      0.13 |       $30.0 \pm 0.10$%      |
> |      0.14 |       $29.9 \pm 0.08$%      |
> |      0.15 |       $29.7 \pm 0.11$%      |
>
> We report the mean and 95% confidence intervals over 10 independent runs and for this controlled experiment, we fix $\alpha=1.0$.

---

> > ### Comment · Reviewer_6c8K · 2022-08-04
> > **Keep rating 7**
> >
> > Thanks for your response. It has solved my question. I would like to keep rating 7.

---

### Official Review · Reviewer_cq35 · 2022-07-06

**Rating:** 7
**Confidence:** 4
**Soundness:** 3 good
**Presentation:** 4 excellent
**Contribution:** 2 fair

**Summary:**

The paper proposes RCAD, a simple regularization scheme for predictive models that aims to increase the predictive entropy for OOD points. OOD points are generated adversarially using FGSM with a large step size. The authors demonstrate that the method reliably improves the test accuracy on a set of image classification tasks, as well as tabular regression tasks. The improvement is more significant when less data is available. To understand why the method works, the authors study it on a toy linear classification task.

**Questions:**

- Does the "sweet spot" for the $\alpha$ value that we see in Figure 4b change significantly with the dataset? Would be useful to understand how tricky it is to get this parameter "right" for a novel task.
- Have authors considered the relation of the method to max margin classifiers like the SVM? I wonder if the two approaches recover similar decision boundaries. A comparison on the toy dataset would be interesting.

**Limitations:**

Authors discuss the impact of the additional regularization term on training time, as well as the assumptions made by the analysis. I consider the potential negative societal impact of this work to be limited.

**Strengths And Weaknesses:**

Strengths:
- The method is clearly introduced, and carefully compared to prior work.
- Statistically significant improvement in test accuracy (up to 3 percentage points) across the benchmarks (image classification and regression on UCI data) when RCAD is employed, compared to multiple strong baselines.
- Ablations to understand the impact of the adversarial step size and model architecture.
- Theoretical analysis for the linear case that motivates the method.

Weaknesses:
- The idea to use adversarial optimization to find OOD points is of limited novelty (see \[a\]). To the best of my knowledge, however, it has not been previously studied theoretically, or evaluated on such a broad set of benchmarks.
- The analysis on the linear case is sometimes hard to follow. Consider moving more detail into the appendix, to give the narrative more breathing space.

References:
- \[a\]: Besnier, V., Bursuc, A., Picard, D., & Briot, A. (2021). Triggering Failures: Out-Of-Distribution detection by learning from local adversarial attacks in Semantic Segmentation. In Proceedings of the IEEE/CVF International Conference on Computer Vision (pp. 15701-15710).

---

> ### Author Response · Authors · 2022-08-02
> **Response to Reviewer cq35**
>
> We thank the reviewer for the detailed feedback and comments. We respond to their two questions individually, in addition to the comment on using adversarial examples as OOD points. As suggested by the reviewer, to improve the readability of the theory section we will move some of the intermediate lemmas in Sec 5 to the Appendix.
>
> > “Sweet spot” for the value of $\alpha$
>
> We find that the optimal choice of $\alpha$ depends more on the size of the training data than the choice of the dataset. For low-data regimes (like CIFAR-100-2k), a higher value of $\alpha=1.0$ is needed to exacerbate the noisy features. When the training data size is large enough (e.g., CIFAR-100, Tiny Imagenet) a smaller value of $\alpha=0.5$ performs better. Note that $\alpha=0.5$ gives good results on all large datasets (Appendix B, L962). To study this further, we furnish results in Fig 3b over two additional datasets in the high-data regime: CIFAR-100 and Tiny Imagenet, where we find that the optimal $\alpha$ is close to $0.5$. We show the mean and 95% confidence intervals over 5 independent runs.
>
> | $\alpha$ | Test Accuracy on CIFAR-100 | Test Accuracy on Tiny Imagenet |
> |---------:|:--------------------------:|:------------------------------:|
> |      0.1 |     $72.4 \pm 0.07 $%     |       $63.8 \pm 0.08 $%       |
> |      0.2 |     $75.2 \pm 0.08 %$%     |       $64.1 \pm 0.10 $%       |
> |      0.3 |     $74.9 \pm 0.07 %$%    |       $64.0 \pm 0.11 $%       |
> |      0.4 |     $77.4 \pm 0.09 %$%     |       $67.0 \pm 0.07 $%       |
> |      0.5 |     $77.3 \pm 0.07 %$%    |       $67.3 \pm 0.09 $%    |
> |      0.6 |     $76.9 \pm 0.09 %$%     |       $67.5 \pm 0.11 $%      |
> |      0.7 |     $77.0 \pm 0.08 %$%     |       $67.2 \pm 0.10 $%       |
> |      0.8 |     $76.8 \pm 0.07 %$%     |       $67.0 \pm 0.10 $%       |
> |      0.9 |     $76.6 \pm 0.08 %$%     |       $66.7 \pm 0.08 $%       |
>
> > Relation to max-margin classifiers
>
> While in some settings the classifier learned by RCAD is well aligned with the max-margin classifier, in other settings like our analysis in Sec 5 the ERM solution converges to the max-margin classifier that depends on the spurious feature while RCAD recovers a classifier with a smaller dependence on the spurious feature (see Fig 5a).
>
> Soudry et al. [46] show that homogenous linear predictors trained with SGD converge to the direction of the max-margin classifiers on linearly separable datasets. In our toy setting, the training data is indeed linearly separable (Fig 5a). Additionally, we train ERM using a margin based loss (L282). Thus, ERM itself recovers the max-margin classifier on the 9 dimensional input data (same as the one returned by SVM solver). However, note that here the max-margin solution relies on the spurious features $x_2, \ldots, x_9$ to maximize the margin. So the SVM solver would fail to recover the true generalizing solution of  $\mathbf{w}^* = [1, 0, \ldots, 0]^\top$, which has a smaller margin. RCAD uses the self-generated examples to identify the spurious features that lie along uncorrelated directions and maximizing entropy on these examples biases the solution of RCAD away from the poorly generalizing max-margin solution, towards the optimal $\mathbf{w}^*$.
>
>
> > Using adversarial examples as OOD points (Besnier et al.)
>
> While adversarial examples have indeed been applied in many applications [11, 18], we believe that ours is the first work to maximize a model’s uncertainty on heavily perturbed adversarial examples, in a way that improves in-distribution test performance. For comparison, Besnier et al. use adversarial examples in a different way, as proxy OOD examples for the OOD detection module. While we generate the adversarial examples with large perturbations, making them out-of-distribution, we instead use them to amplify spurious correlations that need to be unlearnt, as opposed to improving uncertainty quantification or detection under test time distribution shifts.

---

> > ### Comment · Reviewer_cq35 · 2022-08-05
> > **Re: Response to Reviewer cq35**
> >
> > I thank the authors for their response and revisions. After reading the authors response and other reviews, I maintain my rating. I agree with the authors that the way the generated OOD points are _used_ is different in Besnier et al.. At the same time, I encourage the authors to give credit to this earlier work for introducing the adversarial method for _generating_ OOD points. Standard adversarial training methods, on the hand, aim to generate _in-distribution_ points, effectively augmenting the training set.

---

> > > ### Author Response · Authors · 2022-08-05
> > > **Discussion on Besnier et al.**
> > >
> > > Thanks for the additional feedback! We have revised the related work section (L83) to highlight how Besnier successfully uses adversarial examples for OOD detection. **Please let us know if there are additional questions or concerns that we can address**.

---

### Official Review · Reviewer_uiAQ · 2022-07-08

**Rating:** 6
**Confidence:** 4
**Soundness:** 3 good
**Presentation:** 3 good
**Contribution:** 3 good

**Summary:**

This paper proposes a new regularization technique for deep neural networks by reducing confidence on self-generated examples that lie along directions adversarially chosen to increase training loss. The proposed regularizer can be easily integrated into training pipelines and added to existing techniques to gain 1-3% improvement in test accuracy.

**Questions:**

1. Justify the way to generate out-of-distirbution examples.
2. Check how vulnerable the trained model is to adversarial attacks.
3. Discuss the difference from contrastive learning with data augmentation.
4. Running time comparison.
5. Comparison with existing regularizers.

**Ethics Review Area:**

["I don’t know"]

**Limitations:**

Yes.

**Strengths And Weaknesses:**

Strength:

This paper identifies a new inductive bias to reduce the confidence on out-of-distribution inputs and turns this bias into a new regularizer for training. Using adversarial direction to guide the training process is an interesting idea. The empirical evidence demonstrates the effectiveness of the proposed method, and the theoretical analysis also presents some theoretical guarantees on generalization gap. Some interesting insights are also provided with experimental validations.

Weaknesses:

1. The approach of generating out-of-distribution examples from large stepsize adversarial examples needs to be justified. Why is it a proper way to generate such examples? How is it compared with other generation methods, e.g., random samples in other classes?

2. It would be better to point out that the objective is not to improve adversarial robustness although adversarial examples are employed for training. This could help avoid confusion. Although the proposed method does not try to robustfy the model, it is unclear if the proposed method improves the test accuracy at the cost of the adversarial robustness. Corresponding experiments could be designed to address this concern.

3. The connection of the proposed method to some other ideas such as data augmentation and contrastive learning could be also discussed and compared if relevant.

4. Running time comparison could be added to see how much cost paid for the performance gains.

5. It would be also interesting to see how the proposed regularizer is compared with the existing ones with respect to generalization performance.

---

> ### Author Response · Authors · 2022-08-02
> **Response to Reviewer uiAQ**
>
> We thank the reviewer for the detailed feedback. As we discuss below, results on adversarial robustness of RCAD and comparison with existing regularizers is already present in our submission, and we are adding clarifications on running time comparisons. Please let us know if our revisions and clarifications address all of the major issues, or if there are any other concerns. We look forward to continuing the discussion.
>
> > Justify the way to generate out-of-distribution (OOD) examples.
>
> We generate out-of-distribution examples via large adversarial perturbations because: i) empirically we found it to work better than alternative approaches; and ii) it is supported by our theoretical findings.
>
> [Alternative approaches] In our initial experiments, we tried to maximize entropy on OOD examples generated in two different ways: i) random samples from a uniform prior over 3-channel images $[-1, 1]^3$; and ii) interpolating examples from same/different classes. The first approach made no difference in test accuracy. The second is similar to Mix-MaxEnt [36] (see our discussion in Sec 2, L100). While their self-generation logic is geared towards improving *out-of-distribution* uncertainty quantification, ours is focused on improving *in-distribution* test performance. The reviewer’s suggestion of using random samples from other classes is similar to a special case of RCAD when we set $\alpha=0.$, and as we see in Fig 4b this choice of $\alpha$ is sub-optimal.
>
> [Theoretical findings] We generate OOD examples by taking *large adversarial* steps for two reasons: i) adversarial examples amplify spuriously learnt noisy features [18]; and ii) maximizing entropy on all examples that contain a specific feature would prevent the model from using that feature for prediction. We detail this logic in Sec 3 (L153) and verify it theoretically in Sec 5. With regards to the need for the large step-size, note that in Theorem 5.4 we show that unlearning happens when $\alpha$ is large enough, tracking our empirical findings in Fig 4b (L247).
>
>
> > Check how vulnerable the trained model is to adversarial attacks.
>
> Our original submission already had experiments verifying the adversarial robustness of RCAD to FGSM attacks in Appendix C.1 (Tab 6). We find that RCAD improves test performance without decreasing adversarial robustness. We compare RCAD’s adversarial robustness with ADA, ME-ADA and standard adversarial training, all of which explicitly robustify the model’s predictions to FGSM attacks.
>
> On CIFAR-100-2k (low data regime), we find that RCAD not only improves over all baselines by $\geq 1$% on unperturbed test set (clean), it also has the best performance on the test examples adversarially perturbed with FGSM attacks by $\geq 0.5$%. Indeed, on CIFAR-10, CIFAR-100 (high data regime) we find that while RCAD still has the best test accuracy on clean data, ME-ADA has better performance on adversarial test inputs. But RCAD certainly improves over ERM’s test performance on adversarial examples by about $4-6$% on both these datasets.
>
> As suggested, we will make it more clear in the introduction that RCAD’s main objective is to improve test performance, not adversarial robustness even though it employs adversarial examples for it.
>
>
> > Discuss the difference from contrastive learning with data augmentation.
>
> We thank the reviewer for the interesting connection and suggestion on comparing our method with ideas from the self-supervised literature like contrastive learning on image augmentations. We believe that while both RCAD and contrastive learning with data augmentation aim to improve the quality of the learned features, RCAD does so without requiring data augmentation, which often requires domain knowledge to perform. This is especially highlighted in our results on regression tasks (Fig 4c, L252) where we seamlessly apply our RCAD objective without any specific knowledge of the underlying task.
>
> > Running time comparison.
>
> RCAD increases the running time by 30% compared to ERM (L367), which is much faster than alternative adversarial data augmentation baselines ADA and ME-ADA which increase running time by 160% and 185% (over ERM), respectively. We add discussion on this in Appendix B.
>
> > Generalization performance comparison with existing regularizers.
>
> Our main paper already studies how RCAD improves test accuracy in comparison with and in addition to other regularization methods (Tab 2). We compare RCAD with standard data augmentation, label smoothing, advanced augmentation methods like CutOut/CutMix and MixUp training. Please let us know if you feel there is any particularly effective regularizer that would further improve our study.

---

> > ### Author Response · Authors · 2022-08-06
> > **Are there any additional questions or concerns?**
> >
> > We hope that our answers in the author response and the running time comparisons we added have addressed the concerns raised in the review. **We would be happy to continue the discussion if the reviewer has additional questions or concerns.**

---

> > > ### Author Response · Authors · 2022-08-09
> > > **Has our response addressed the concerns?**
> > >
> > > Dear Reviewer,
> > >
> > > Thank you for the suggestions for improving the paper. As we mention in our response, we already have in our submission experimental results on the adversarial robustness of models trained with our objective (RCAD). We also point to a section of our paper that justifies RCAD's method of generating OOD examples. We hope that **this addressed your concerns**. If not, we would be happy to continue the discussion and/or revise the paper.

---

### Official Review · Reviewer_wyGm · 2022-07-11

**Rating:** 6
**Confidence:** 3
**Soundness:** 2 fair
**Presentation:** 3 good
**Contribution:** 2 fair

**Summary:**

This paper proposes a regularization method for improving the generalization in maximum likelihood learning. The main idea is to reduce confidence (or increase entropy) on inputs generated by large adversarial perturbation. The authors conduct extensive experiments to demonstrate the effectiveness of their method. They also analyze the effectiveness of their method by showing it can unlearn noisy features in a toy setting.

**Questions:**

The authors compare their method with baselines including those from adversarial training such as FGSM, and those from domain adaptation such as ADA and ME-ADA. But as far as I am concerned, none of these methods is specifically designed for standard in-domain generalization. FGSM is for improving adversarial robustness while ADA and ME-ADA are for domain generalization. Why didn't the authors compare their method to VAT [1], which also proposes to perform data augmentation with adversarial perturbation, and aims for generalization in standard supervised and semi-supervised settings?

[1] Virtual adversarial training: a regularization method for supervised and semi-supervised learning. Miyato et al., 2018.

**Limitations:**

The limitations have been properly discussed in Section 6.

**Strengths And Weaknesses:**

This paper is overall well-organized and clearly written. As the main contribution of this paper is to propose a new regularization method, the novelty might be minor since the proposed method resembles the existing methods in adversarial data augmentation [1,2,3], albeit the difference that this paper proposes to "unlearn" examples under large adversarial perturbation instead of "learn" examples under small adversarial perturbation.

My biggest concern in this paper is that the experiment results may not be sufficient to demonstrate the effectiveness of the proposed method. Since the authors aim to improve generalization in a very general sense instead of robustness against perturbation or domain shift, experiments on large datasets such as ImageNet would be necessary. I am not quite certain about the effectiveness of the proposed method in such a more realistic setting as the improvements on full datasets such as CIFAR-10 and CIFAR-100 are already marginal. The authors also emphasize that the proposed method is most effective in low-data regimes. In this regard, I believe it would be necessary to showcase the proposed method in a semi-supervised setting with unlabeled data available.

[1] Virtual adversarial training: a regularization method for supervised and semi-supervised learning. Miyato et al., 2018.

[2] Generalizing to unseen domains via adversarial data augmentation. Volpi et al., 2018.

[3] Maximum-entropy adversarial data augmentation for improved generalization and robustness. Zhao et al., 2020.

---

> ### Author Response · Authors · 2022-08-02
> **Response to Reviewer wyGm**
>
> We thank the reviewer for the detailed feedback and comments. The two main concerns seem to be: i) experimental results on more realistic benchmarks like ImageNet; and ii) comparison with semi-supervised methods like VAT in low-data regimes. As we discuss below, we already included experiments on ImageNet in the submission, and we are also adding new comparisons to VAT. Please let us know if our revisions and clarifications address all of the major issues, or if there are any other concerns. We look forward to continuing the discussion.
>
> > Imagenet Results
>
> The original submission had results on Imagenet (Appendix B, Tab 5), finding that RCAD improves the top-1 accuracy by 0.31%. We are not surprised that the gains are smaller in this high-data regime since the relative improvement from RCAD is more pronounced when training data is limited (Sec 4.3, Fig 3b). Nevertheless, because the Imagenet benchmark is competitive, even small, statistically significant benefits have been recognized as important contributions [AR1, AR2, AR3].
>
> [AR1] Xie, Qizhe, et al. "Unsupervised data augmentation for consistency training." Advances in Neural Information Processing Systems 33 (2020): 6256-6268.
>
> [AR2] Tsipras, Dimitris, et al. "From Imagenet to image classification: Contextualizing progress on benchmarks." International Conference on Machine Learning. PMLR, 2020.
>
> [AR3] List of pre-trained models on ImageNet along with validation accuracies: https://github.com/Cadene/pretrained-models.pytorch
>
> > Comparisons with semi-supervised baseline: Virtual Adversarial Training (VAT)
>
> As suggested by the reviewer, we adapted RCAD to the semi-supervised setting and compared it to VAT, finding that RCAD improves over VAT by 0.29% and 0.17% on CIFAR-100-2k and CIFAR-100-10k respectively (see table below). We have added this result and corresponding discussion to Appendix E.
>
> We run VAT on the subsampled CIFAR training sets: CIFAR-100-2k and CIFAR-100-10k with 2k and 10k training samples respectively. The rest of the CIFAR-100 training data is treat as unlabeled data on which VAT performs adversarial training treating the trained model’s predicted label as the true one. Analogous to this, the semi-supervised version of our objective RCAD optimizes the objective in Eqn. 2 (L144) on the labeled set $\hat{\mathcal{D}_l}$, but for each unlabeled example (in $\hat{\mathcal{D}_u}$) we simply maximize entropy on its corresponding OOD example generated using Eqn. 1 (L141) with the main difference being that we treat the model’s predicted label $\hat y$ as the true label (similar to VAT). Thus, the semi-supervised RCAD objective is given by:
>
>
> $
> \arg\max_{\mathbf{w} \in \mathcal{W}} \mathbb{E}_\hat{\mathcal{D}_l} [ \log p_\mathbf{w} (y \mid \mathbf{x}) + \lambda \cdot {\mathcal{H}}_\mathbf{w} ( \mathbf{x} - \alpha \cdot {\nabla}_\mathbf{x}  \log p_\mathbf{w} (y \mid \mathbf{x}) ) ] +  \lambda \cdot \mathbb{E}_\hat{\mathcal{D}_u} [ {\mathcal{H}}_\mathbf{w} ( \mathbf{x} - \alpha \cdot \nabla_\mathbf{x}  \log p_\mathbf{w} (\hat y \mid \mathbf{x}) ) ]
> $
>
> where, $
> \hat y = \arg\max_{y \in \mathcal{Y}} p_{\mathbf{w}}(y \mid \mathbf{x})
> $
>
>
> |                      |    CIFAR-100-2k    |    CIFAR-100-10k   |
> |---------------------:|:------------------:|:------------------:|
> |                  VAT |  32.74 $\pm$ 0.06% | 62.81 $\pm$ 0.06% |
> | semi-supervised RCAD | 33.03 $\pm$ 0.05% | 62.98 $\pm$ 0.05% |
>
>
>
> The $95$% confidence intervals are obtained by evaluating the test performance of models on 10 independent runs for both methods. We trained VAT using the original paper authors’ implementation and set hyperparameters $\alpha, \epsilon$ (see Eqns. 5, 8 in Miyato et al.)  to $\alpha=1.0, \epsilon=8.0$ for both datasets. RCAD hyperparameters were set to $\alpha=0.7, \lambda=0.1$ for CIFAR-100-2k and $\alpha=0.5, \lambda=0.02$ for CIFAR-100-10k. For both methods, the respective hyperparameters were tuned using a hold out validation set
>
> > Proposed method resembles the existing methods in adversarial data augmentation
>
> We are unaware of any method that improves test performance by maximizing uncertainty on generated examples, whether those examples are generated randomly, through data augmentation, or via adversarial perturbations (adversarial data augmentation). Adversarial data augmentation methods [1,2,3] *minimize* uncertainty (entropy) on the self-generated examples, whereas RCAD *maximizes* uncertainty. Our experimental results confirm that RCAD outperforms these prior methods that minimize uncertainty (Fig 3a).

---

> > ### Author Response · Authors · 2022-08-06
> > **Are there any additional questions or concerns?**
> >
> > We hope that the Imagenet results, additional experiments on VAT,  and comparisons with adversarial data augmentation methods have addressed the concerns raised in the review. **We would be happy to continue the discussion if the reviewer has additional questions or concerns.**

---

> > > ### Author Response · Authors · 2022-08-09
> > > **Have the new experiments addressed the concerns?**
> > >
> > > Dear Reviewer,
> > >
> > > Thank you for the suggestions for improving the paper. We have additional experiments comparing RCAD with the baseline VAT in a semi-supervised setting. As mentioned in our response, we have edited our submission with these experiments that we believe further strengthen the paper. Together with the discussion on Imagenet results and comparisons with adversarial data augmentation methods, **have all the concerns been addressed?** If not, we would be happy to continue the discussion and/or revise the paper.

---

> > > ### Comment · Reviewer_wyGm · 2022-08-09
> > > **Thank you for your reply**
> > >
> > > The additional results on VAT partly address my concern and I have increased my score. But the effectiveness of the proposed method still worries me given the marginal improvement.

---

### Meta-Review · Area_Chair_UfYN · 2022-08-26

**Recommendation:** Accept
**Confidence:** Certain

**Metareview:**

All reviewers have expressed a clear opinion in favour of acceptance, one improving their score after the rebuttal and discussion. I’m happy to recomment acceptance.

**Award:**

No

---

### Decision · Program_Chairs · 2022-09-14

Accept